# Uncovering the basis of protein-protein interaction specificity with a combinatorially complete library

Thuy-Lan V Lite[1], Robert A Grant[1], Isabel Nocedal[1], Megan L Littlehale[1], Monica S Guo[1], Michael T Laub[1,2]*

[1]Department of Biology Massachusetts Institute of Technology, Cambridge, United States; [2]Howard Hughes Medical Institute Massachusetts Institute of Technology, Cambridge, United States

**Abstract** Protein-protein interaction specificity is often encoded at the primary sequence level. However, the contributions of individual residues to specificity are usually poorly understood and often obscured by mutational robustness, sequence degeneracy, and epistasis. Using bacterial toxin-antitoxin systems as a model, we screened a combinatorially complete library of antitoxin variants at three key positions against two toxins. This library enabled us to measure the effect of individual substitutions on specificity in hundreds of genetic backgrounds. These distributions allow inferences about the general nature of interface residues in promoting specificity. We find that positive and negative contributions to specificity are neither inherently coupled nor mutually exclusive. Further, a wild-type antitoxin appears optimized for specificity as no substitutions improve discrimination between cognate and non-cognate partners. By comparing crystal structures of paralogous complexes, we provide a rationale for our observations. Collectively, this work provides a generalizable approach to understanding the logic of molecular recognition.

## Introduction

Protein-protein interactions underlie most cellular processes and are the basis of established and emerging therapies such as monoclonal antibodies (*Weiner et al., 2010*), chimeric antigen receptor T cells (*Brentjens et al., 2013*), and stapled peptide drugs (*Walensky and Bird, 2014*). To prevent non-specific and potentially detrimental interactions, proteins must discriminate between cognate and non-cognate interaction partners. This is often achieved via molecular recognition or non-covalent interactions between protein surfaces. However, the structural space of interfaces is degenerate, with nearly 90% of native interfaces having a close structural neighbor with a highly similar backbone geometry (*Gao and Skolnick, 2010*). This problem is particularly acute for cells that encode paralogous protein families, whose members can share significant structural and sequence similarity. Prior work has shown that the specificity of many paralogous protein families is determined by a subset of residues that typically map to the interface formed by a given protein and its binding partner(s) (*Aakre et al., 2015*; *Ovchinnikov et al., 2014*; *Skerker et al., 2008*).

Even when the specificity residues governing a given type of protein-protein interaction have been identified, it remains unclear how individual residues contribute to enforcing specificity. Collectively, specificity-determining residues must achieve two goals: (i) promote or stabilize the desired, cognate interaction and (ii) inhibit or destabilize all unwanted, non-cognate interactions, which we call positive and negative specificity elements, respectively (*Schreiber and Keating, 2011*). If a residue functions strictly as a positive element, substitutions at this site will destabilize the cognate interaction, but not affect non-cognate interactions (*Figure 1A*). Conversely, if a residue is strictly a negative element, substitutions will not attenuate the cognate interaction, but could produce cross-

*For correspondence:
laub@mit.edu

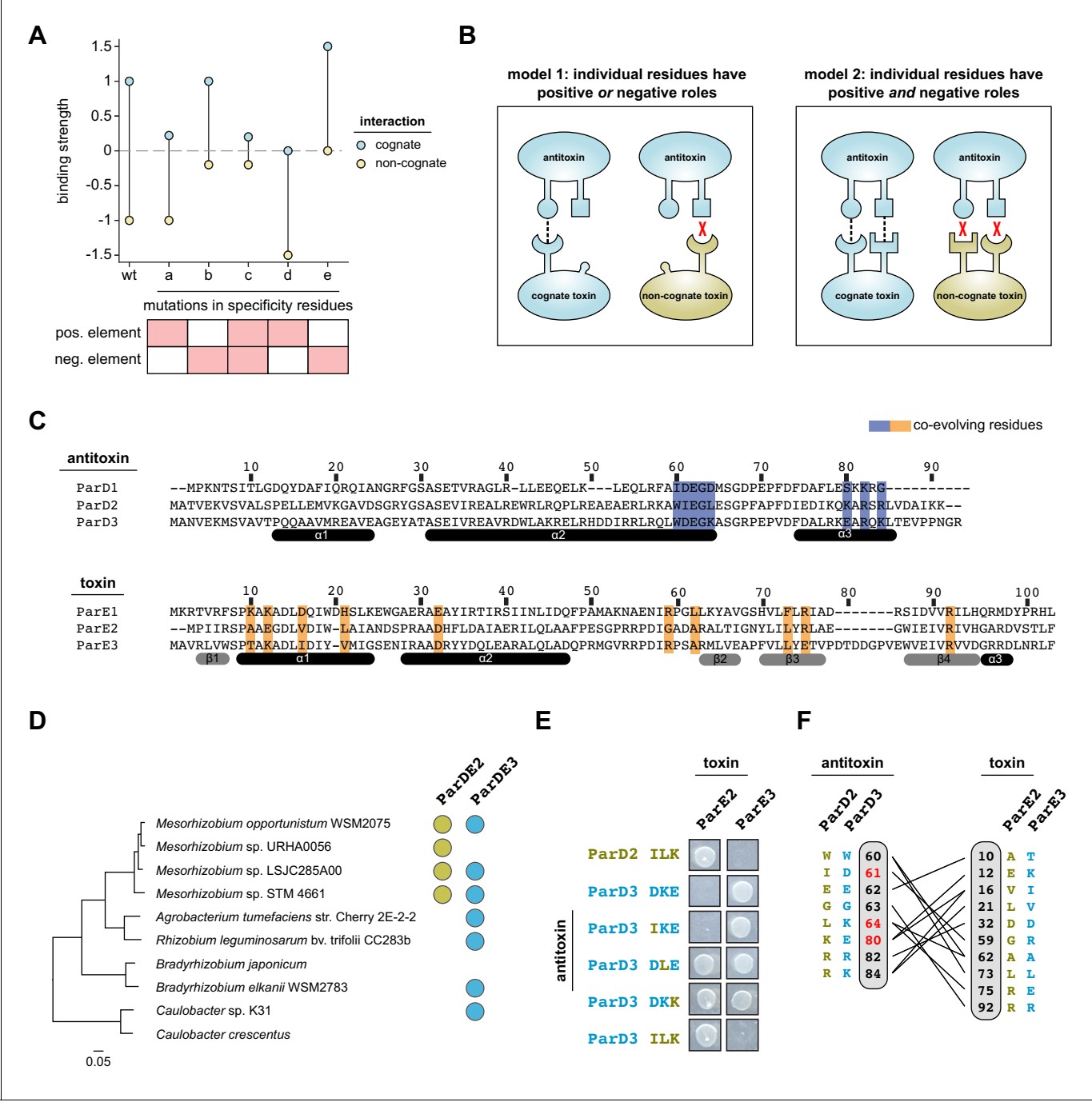

**Figure 1.** Specificity-determining residues that dictate toxin-antitoxin interactions. (**A**) Schematic representing how antitoxin mutations in specificity-determining residues might affect binding to the cognate toxin (blue) or the non-cognate toxin (gold). A wider gap in binding preference reflects greater specificity. The effects of mutating a specificity residue that serves a positive specificity role (**a**), a negative specificity role (**b**), or both roles (**c**) are shown. Mutations that represent possible trade-offs between binding the cognate toxin and discriminating against the non-cognate toxin are also shown (**d–e**). (**B**) Models for how positive and negative specificity determinants are distributed across the interface. Model 1: positive and negative design are accomplished by distinct interface residues. Model 2: individual residues can serve both roles. (**C**) Sequence alignment of three ParD-ParE systems from *Mesorhizobium opportunistum*. Coevolving residues are highlighted in purple (ParD) or orange (ParE). (**D**) Phylogenetic tree inferred from protein sequences of 15 highly conserved genes. Distribution of ParDE2 systems and ParDE3 systems are indicated in gold and blue, respectively. (**E**) Toxicity-rescue assay for wild-type ParD2, wild-type ParD3, or the ParD3 variant indicated, each co-expressed with either wild-type ParE2 or ParE3 toxin. The ParD3 variants harbor subsets, as noted, of the mutations D61I, K64L, and E80K. (**F**) Residues in ParD and ParE that strongly coevolve (probability score >0.95) with lines connecting covarying pairs. Residues are numbered according to their position in the alignments in panel (**C**). Positions selected for the ParD3 saturation mutagenesis library are indicated in red.

*Figure 1 continued on next page*

*Figure 1 continued*

The online version of this article includes the following figure supplement(s) for figure 1:

**Figure supplement 1.** ParE protein tree.

---

talk with non-cognate partners. For most protein complexes, it is not clear if individual residues serve primarily as positive or negative elements, or both (*Figure 1A–B*). Additionally, it is not clear if specificity residues are optimized for either role, or whether trade-offs arise that prevent optimization. Understanding how individual residues contribute to specificity is important for rationally engineering protein interaction specificity (*Chen and Keating, 2012*) and for understanding how paralogous protein complexes evolve (*McClune and Laub, 2020*).

Dissecting the roles of interface residues has typically involved analyzing limited numbers of substitutions. A common strategy is to replace interface residues with alanines (*Ashkenazi et al., 1990*; *Clackson and Wells, 1995*; *Kristensen et al., 1997*; *Zhang and Palzkill, 2004*) or the corresponding residues from a homologous protein (*Brasch et al., 2018*; *Bridgham et al., 2006*; *Cosmanescu et al., 2018*; *Sergeeva et al., 2020*; *Stiffler et al., 2007*) and assay the effects on interaction specificity. In some cases, all possible single substitutions have been introduced, either at key positions or, more recently, at all positions within a protein in an approach called 'deep mutational scanning' (*Fowler et al., 2010*; *Fowler and Fields, 2014*; *McLaughlin et al., 2012*; *Whitehead et al., 2012*). However, single substitutions may not reveal the full impact of a given residue because the effect of a mutation can depend strongly on the context, or genetic background, in which it is introduced (*Melamed et al., 2013*; *Natarajan et al., 2013*; *Podgornaia and Laub, 2013*; *Starita et al., 2013*).

Considering the effect of a substitution in multiple genetic contexts—in other words, considering combinations of substitutions—can better illuminate the contributions of interface residues to specificity (*Dutta et al., 2010*; *Jenson et al., 2018*; *Salinas and Ranganathan, 2018*). Many studies have leveraged large, combinatorial libraries to identify protein variants that bind the desired target such as bacterial Cry toxins (*Dong et al., 2020*; *Jiao et al., 2017*), HIV antigens (*Barbas et al., 1994*; *Burton et al., 1991*), or cytokine receptors (*Fairlie et al., 2004*). However, these studies did not systematically examine the effects of substitutions at a given protein interface. Other combinatorial library studies have systematically dissected protein interactions but focused on a single interacting pair such as the proto-oncogenes Fos and Jun (*Diss and Lehner, 2018*), PDZ domains and their cognate ligands (*Salinas and Ranganathan, 2018*), and the two-component signaling proteins PhoP-PhoQ (*Podgornaia and Laub, 2013*). These studies did not consider non-cognate interactions, precluding conclusions about interaction specificity. Finally, other work has leveraged combinatorial libraries to identify the determinants of specificity for a cognate partner versus a closely-related, non-cognate partner in the context of Bcl-2 family proteins and their peptide ligands (*Dutta et al., 2010*; *Jenson et al., 2017*; *Jenson et al., 2018*) and for bacterial toxin-antitoxin systems (*Aakre et al., 2015*). However, these studies did not consider all combinations of substitutions. In sum, prior work has not systematically dissected protein interaction specificity by examining the effects of substitutions in a combinatorially complete manner.

Here we built and analyzed libraries containing all 8000 possible combinations of three key specificity residues of a protein complex. Our work focuses on the ParD-ParE family of toxin-antitoxin (TA) systems, which are often found on bacterial plasmids and chromosomes (*Fraikin et al., 2020*). Most TA systems include a stable toxin that can inhibit cell growth but that is normally bound and inhibited by a cognate antitoxin. ParE toxins likely restrict cell growth by targeting topoisomerases (*Jiang et al., 2002*; *Yuan et al., 2010*). Because ParE is inhibited by a cognate ParD antitoxin, cell growth is a convenient readout for binding. TA systems are typically co-operonic, and new paralogs frequently arise through operon duplication. Importantly, these paralogous systems are highly specific, meaning antitoxins typically inhibit only their cognate, co-operonic toxin (*Aakre et al., 2015*; *Fiebig et al., 2010*).

For two insulated ParD-ParE systems derived from a recent duplication, we systematically probed the contributions of three key residues to specificity by generating a combinatorially complete library at these three positions in the antitoxin ParD3. Using a bulk competition assay coupled with deep sequencing, we identified the degree to which all variants antagonize the cognate toxin (ParE3) and

the non-cognate toxin (ParE2). We not only uncovered the effects of every possible single mutation in the wild-type background but a complete distribution of effects for that mutation in hundreds of genetic backgrounds. This approach reveals the fundamental contributions of a given residue to specificity, independent of genetic context. We find that the three interface positions are each positive elements that promote the cognate interaction, but only two positions serve as negative elements that block the non-cognate interaction. Thus, positive and negative contributions are neither inherently coupled nor mutually exclusive. Notably, when considering the full distribution of effects in hundreds of genetic backgrounds, we find no substitutions that improve the discrimination of cognate and non-cognate partners. These findings suggest that the residues in wild-type ParD3 are optimal with respect to promoting the insulation of these two ParD-ParE systems. By solving a co-crystal structure of ParD2-ParE2 and comparing it to an existing structure of ParD3-ParE3, we identify plausible mechanistic explanations for how each residue contributes to specificity. Altogether, our work provides a comprehensive library-based approach for systematically dissecting protein interaction specificity and a framework for understanding how paralogous systems avoid unwanted cross-talk.

## Results

### Recently-duplicated ParD-ParE systems use a minimal essential set of specificity residues

To examine insulation between paralogous toxin-antitoxin loci, we focused on two closely-related systems from the widespread ParD-ParE family. Although the first ParD-ParE system characterized was plasmid-borne (*Roberts and Helinski, 1992*), these TA systems are often found in multiple copies on bacterial chromosomes (*Aakre et al., 2015*; *Leplae et al., 2011*). We previously showed that ParD antitoxins are highly specific for their cognate ParE toxins, and that interaction specificity is controlled by a discrete set of coevolving residues that map to the protein-protein interface formed by ParD and ParE (*Aakre et al., 2015*). To further refine the set of residues that determine the interaction specificity of ParD and ParE, we searched for coevolving residues using GREMLIN (*Ovchinnikov et al., 2014*), as done previously, using a significantly stricter E-value cut-off ($E < 10^{-20}$). This analysis revealed eight residues in ParD that strongly covary with 10 residues in ParE (*Figure 1C*).

These coevolving residues are likely to determine the interaction specificity of paralogous ParD-ParE systems. However, paralogs derived from a recent duplication event may use only a subset of these residues to maintain insulation. The previously-studied ParD2-ParE2 and ParD3-ParE3 systems from the α-proteobacterium *Mesorhizobium opportunistum* are highly similar, with 41% (antitoxins) and 42% (toxins) identity. To determine whether these systems arose from a recent duplication event, we first built a protein tree of ParE toxins in α-proteobacteria, which revealed that ParE2 and ParE3 are monophyletic and represent sister taxa (*Figure 1—figure supplement 1*; *Source data 1–2*). We then built a species tree for α-proteobacteria using a concatenated alignment of 15 conserved genes and implemented HMMER to identify ParE2 and ParE3 paralogs (*Source data 3–4*). This species tree revealed that ParE3 paralogs were more widely distributed, and that organisms that have a ParE2 ortholog usually carry a ParE3 ortholog (*Figure 1D*). Taken together, the protein and species trees suggest that ParD2-ParE2 and ParD3-ParE3 share a recent common ancestor, and that ParD2-ParE2 is the derived system produced by gene duplication.

Despite their close evolutionary relationship, the two systems do not cross-talk (*Aakre et al., 2015* and *Figure 1E*). Of the 8 ParD specificity-determining residues identified in our coevolution analysis, four are identical between ParD3 and ParD2, and one residue is positively-charged in both (*Figure 1C and F*). Thus, ParD2 harbors only three non-conservative substitutions relative to ParD3 in its specificity-determining residues. Notably, swapping all ParD3 residues for the derived residues in ParD2 at these three positions (i.e. D61I/K64L/E80K, or DKE to ILK) was sufficient to rewire its specificity, producing an interaction with the non-cognate toxin ParE2 and eliminating the native interaction with ParE3 (*Figure 1E*). Replacing just one of these three residues in ParD3 with the ParD2 residue was sometimes sufficient to produce promiscuity (*Figure 1E*). The substitutions K64L and E80K in ParD3 each individually enabled ParD3 to interact with the non-cognate toxin ParE2, while retaining its interaction with ParE3. By contrast, the D61I substitution did not detectably change the interaction specificity of ParD3. Collectively, these results suggest that (i) the coevolving

residues identified by GREMLIN dictate the interaction specificity of ParD-ParE complexes and (ii) these specificity-determining residues make different contributions to preventing cross-interactions between the paralogous systems ParD2-E2 and ParD3-E3.

## Mapping fitness in a saturated interface mutant library

The targeted mutational studies above offer only limited insight into how individual interface positions contribute to the specificity of ParD-ParE interactions. To more systematically dissect the contributions made by each position in ParD3 to promoting an interaction with the cognate partner ParE3 and to excluding interaction with the non-cognate partner ParE2, we generated a saturation mutagenesis library of ParD3 at the three key interface positions (*Figure 2A*, *Figure 2—figure supplement 1A*). The resulting library thus has a theoretical diversity of 8000 variants. This library offers several advantages over the ParD3 library used previously (*Aakre et al., 2015*), which shares only 130 of 8000 (or 1.6%) of the variants examined here. First, we sought residues that specifically mediate insulation of the ParD3-ParE3 and ParD2-ParE2 systems, and thus focused on residues that are not conserved between the two systems. The prior library mutagenized a position (W60) that is identical between the two antitoxins and is essential for ParD3 to bind either toxin. Second, we generated a combinatorially complete library with full randomization at each position, whereas the previous library included only residues commonly seen at each position in native sequences.

First, to assess the ability of each variant in our library to neutralize the cognate toxin, we co-transformed the library into a strain carrying an arabinose-inducible copy of ParE3 on a plasmid (*Figure 2B*). The ParD3 variants are expressed from an IPTG-inducible promoter on a separate plasmid. As expected, expressing ParE3 arrests cell growth in conditions that do not induce the ParD3 library (*Figure 2—figure supplement 1B*). However, co-expressing the ParD3 library with the ParE3 toxin produced robust growth of the culture, similar to conditions in which cells express neither toxin nor antitoxin. This result implies that the library contains a population of ParD3 variants that retain the ability to antagonize ParE3. To identify these variants, we co-expressed the ParD3 library with ParE3 for 10 hr and then deep-sequenced the entire ParD3 gene before and after selection (*Figure 2B*). We obtained reliable measurements (pre-selection reads >25 and post-selection reads >0) for 7882 variants (*Figure 2—figure supplement 1C*). Enrichment of a given variant in this competitive growth assay should reflect its ability to neutralize the ParE3 toxin. We observed large changes in variant frequency over the course of the experiment. For instance, ParD3 DKE, the wild-type variant, was enriched 20-fold in the presence of ParE3. In contrast, the variant ParD3 ILK, which has 3 ParD2-like substitutions, was de-enriched 17-fold, and variants harboring stop codons were de-enriched 42-fold on average.

To quantitatively compare the behavior of individual variants over the course of the experiment, we calculated a raw fitness score for each variant, as previously described (*Aakre et al., 2015*; *van Opijnen et al., 2009*). Briefly, this raw fitness score ($W_{raw}$) reflects the log-fold expansion of each mutant relative to the remaining variants. We then scaled the raw fitness values by setting the mean fitness of variants encoding stop codons to $W = 0$ and the wild-type sequence DKE to $W = 1$. The fitness scores were highly reproducible between biological replicates ($R^2 = 0.96$, *Figure 2—figure supplement 1D*). To validate our approach, we independently cloned four ParD3 variants from our library into an inducible vector and co-expressed each variant with ParE3 (*Figure 2—figure supplement 1E–F*). We found that the fitness values calculated qualitatively agree with the ability of each variant to rescue growth in vivo following toxin expression. Additionally, for the ~130 variants also queried in our previous work (*Aakre et al., 2015*), the relative fitness was highly reproducible between the two studies ($R^2 = 0.96$; *Figure 2—figure supplement 1G*).

## Pervasive degeneracy in toxin binding

Our library selection revealed a substantial pool of variants that antagonize the wild-type ParE3 toxin. At a threshold of $W \geq 0.5$, we identified 1847 variants out of 7882 (or 23.4%) that neutralize ParE3, including 55 single, 835 double, and 956 triple mutants (*Figure 2C*). The most common amino acid at each position was the wild-type residue—D61, K64, and E80 (*Figure 2D*). A similar pattern was also seen at higher fitness thresholds. Nearly all (55/57 = 96%) single substitutions were tolerated except proline substitutions at the first or second positions, variants PKE and DPE (*Figure 2E*). The majority of double mutants (835/1069 = 78.2%) also retained interaction with the

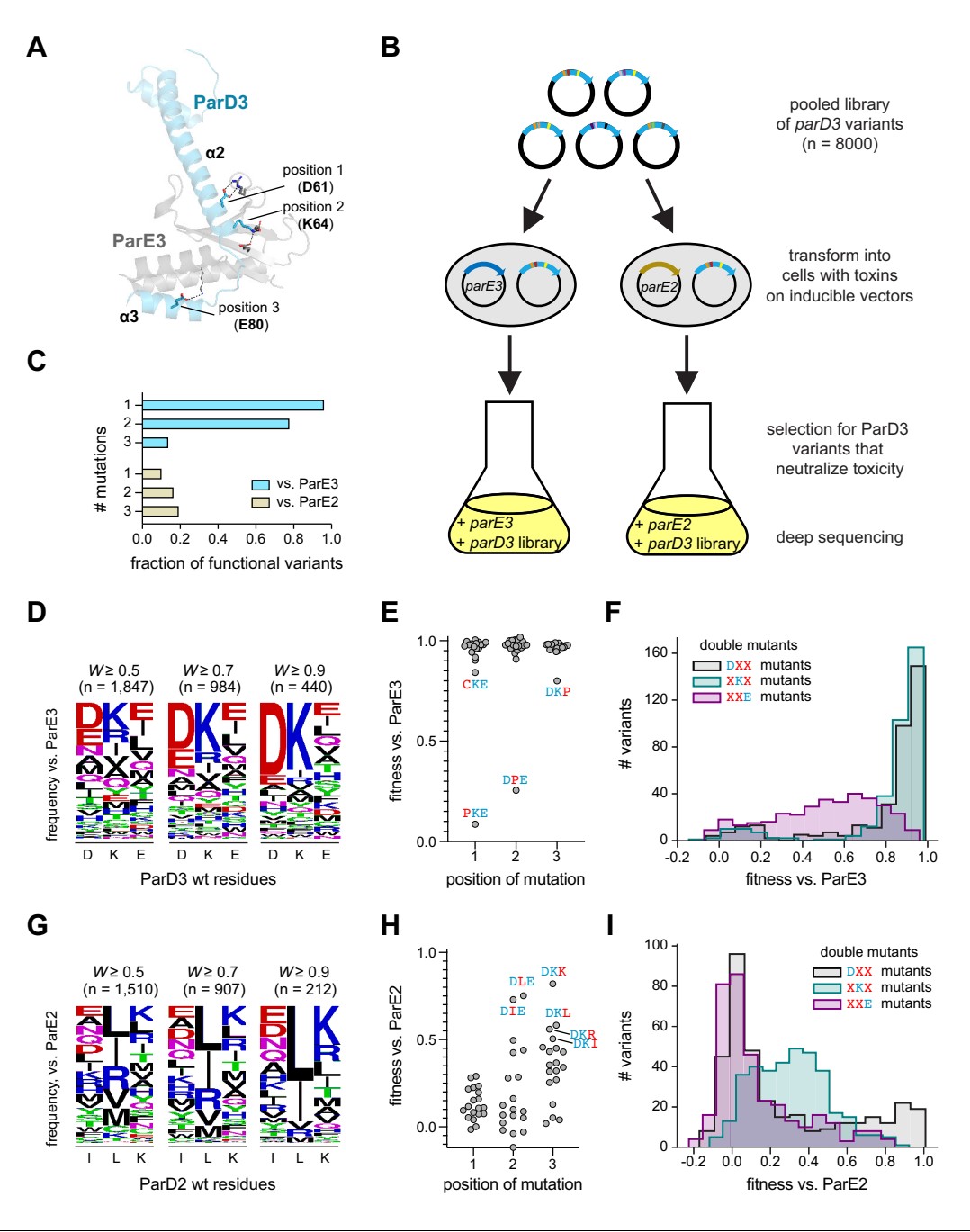

**Figure 2.** Mapping variant fitness via a combinatorially complete library. (**A**) Residues mutated in the saturation mutagenesis library, mapped onto the ParD3-ParE3 crystal structure (PDB: 5CEG). Salt bridges between ParD3 library positions and ParE3 are indicated. Note that ParD3-ParE3 forms a symmetric tetramer with two chains of each protein; for simplicity, only a single chain of each is shown. (**B**) Schematic of the ParD3 library experiment. (**C**) Fraction of single, double, and triple mutants that neutralize ParE3 or ParE2 ($W \geq 0.5$). (**D**) Frequency of amino acids at each position for variants that neutralize ParE3, summarized as a sequence logo, as the fitness thresholds indicated. (**E–F**) Fitness of all single (**E**) and double (**F**) mutants against ParE3. (**G–I**) Same as in (**D**)-(**F**), but for ParE2.

The online version of this article includes the following figure supplement(s) for figure 2:

**Figure supplement 1.** ParD3 library validation and statistics.

**Figure supplement 2.** Comparing the relative fitness of variants across different induction levels.

cognate toxin. Double mutants that retained the wild-type residues D61 (DXX) or K64 (XKX) tended to have higher fitness than double mutants that retained E80 (XXE) (*Figure 2F*), suggesting that position 3 may provide the least binding energy to the native ParD3-ParE3 interaction. Triple mutants, which do not feature any of the wild-type ParD3 residues at the randomized positions, were the least likely to retain an interaction with the cognate toxin (956/6755 = 14.2%), but in absolute numbers comprised the biggest class of functional variants. Taken together, these results show that degeneracy at the ParD3-ParE3 interface is far greater than was previously appreciated (*Aakre et al., 2015*), with nearly a quarter of all variants retaining an ability to antagonize the toxin ParE3.

To identify variants that antagonize ParE2, a non-cognate toxin for ParD3, we repeated the competitive growth experiment but expressed the ParD3 library in cells producing ParE2 (*Figure 2—figure supplement 1H–M*). As before, we calculated a fitness score for each variant, this time scaling the raw values by setting the ParD2-like sequence (ILK) to $W = 1$. The fitness scores were again highly reproducible between biological replicates ($R^2 = 0.95$) and predicted growth in the presence of ParE2 in an independent assay (*Figure 2—figure supplement 1J–K*). We identified 1510 variants that antagonize the ParE2 toxin ($W \geq 0.5$). The antitoxin variants that antagonize ParE2 toxicity often shared properties with the wild-type ParD2 antitoxin (*Figure 2G*). For instance, the most common amino acid at position 2 in the post-selection library was the wild-type ParD2 residue leucine, followed by isoleucine, which has similar chemical properties. Likewise, the most common amino acid at position 3 was the ParD2 residue lysine, followed by arginine, which is also positively-charged. In contrast, at library position 1, residues with a diverse range of structural and physical properties appear in approximately equal frequencies. As with ParE3, similar patterns were seen at higher fitness thresholds (*Figure 2G*).

Variants that improved binding to the non-cognate toxin ParE2 typically had 2 or three mutations relative to the parental, wild-type ParD3 (*Figure 2C*). Only 6 of 57 (10.5%) single mutant variants antagonized ParE2, and, notably, single mutations at position 1 were never sufficient to produce a ParD3-ParE2 interaction (*Figure 2H*). Single mutants that were functional for antagonizing ParE2 either substituted the wild-type ParD3 residue with leucine or isoleucine at position 2 or 3 (DLE, DIE, DKL, DKI) or reversed the charge of the wild-type residue at position 3 (DKK, DKR). Of the double mutants, 180 of 1069 (16.8%) antagonized ParE2; more than half of these (95) harbored mutations in both the second and third library positions while retaining the aspartate found at position 1 in the parental ParD3, that is, DXX double mutants (*Figure 2I*). This result further suggests that positions 2 and 3 are important for inhibiting interaction with the non-cognate toxin. Residues at position 2 that were incompatible with the ParD3-ParE2 interaction include all of the aromatic residues (F, H, Y, W), residues with poor helix-forming propensities (P, G), as well as serine and aspartate. As with ParE3 interactions, we find significant degeneracy, with 1510 variants (19.2% of the library) capable of interacting with ParE2. Our observation of extensive degeneracy was robust to the threshold used for defining productive interactions (*Figure 2G*).

## Library variants show a range of abilities to discriminate between cognate and non-cognate partners

Thus far, we have estimated fitness for library variants against the cognate and non-cognate toxins independently. However, specificity in vivo involves simultaneously binding the correct partner while avoiding unwanted interactions. To visualize the extent to which each variant interacts with both toxins, we generated a scatterplot of ParD3 fitness when screened against each potential partner (*Figure 3A*). This analysis revealed a large pool of variants (666 at a threshold of $W \geq 0.5$) that interact promiscuously with both toxins (*Figure 3B*). This set represents 36% and 44% of total variants that bind ParE3 and ParE2, respectively. For the promiscuous variants, the residues at each position were similar to those that promote interactions with each toxin considered independently (compare *Figure 3C* with *Figure 2D and G*). For instance, the most frequent residues among promiscuous variants at position 3 are leucine and isoleucine. Both amino acids appear frequently at position 3 in the full set of variants that antagonize either toxin.

As expected, the scatterplot showed that the wild-type variant DKE is specific for the cognate toxin ParE3 (*Figure 3A*). Interestingly, we observed a small set of variants with wild-type-like fitness for the cognate toxin (i.e., $W \approx 1$) but with lower fitness against the non-cognate toxin. As depicted in *Figure 1A*, more specific variants will have a wider 'gap' in preference for the cognate versus the

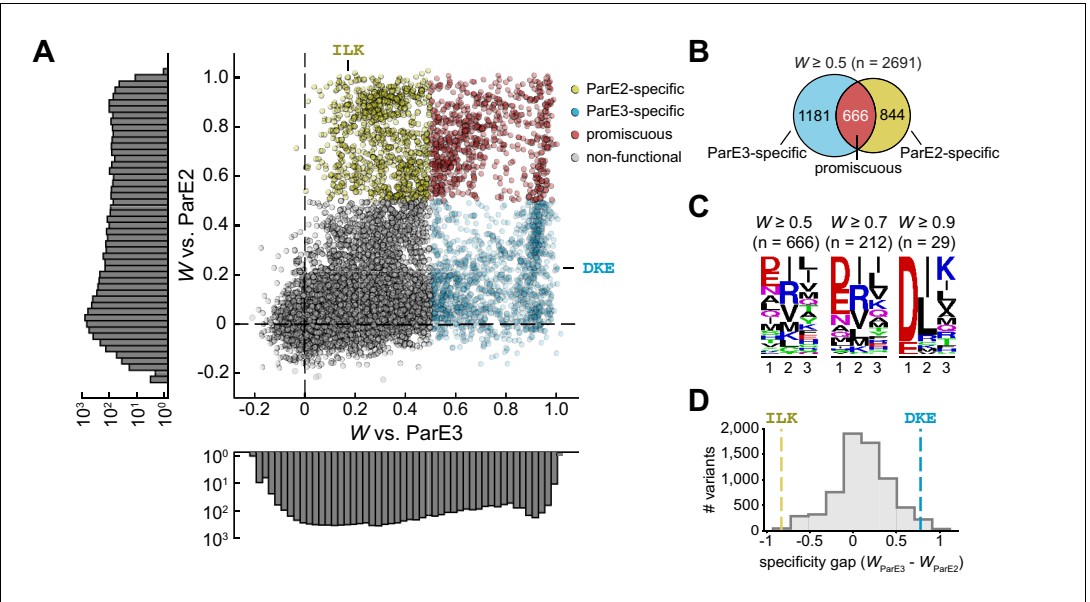

**Figure 3.** The sequence space of antitoxin specificity residues. (**A**) Fitness of ParD3 variants against ParE2 and ParE3. Gold, specific for ParE2; blue, specific for ParE3; red, promiscuous for both toxins. Histograms of fitness values are depicted. (**B**) Venn diagram of ParD3 variants that neutralize ParE3, ParE2, or both. (**C**) Frequency of amino acids at each position for promiscuous variants, summarized as a sequence logo, at the fitness thresholds indicated. (**D**) Specificity gap for library variants, calculated as $W_{ParE3} – W_{ParE2}$. The gap for the wild-type ParD3 variant ('DKE,' blue dashed line) and ParD2-like variant ('ILK,' gold dashed line) are shown. Variants with prolines are omitted.

The online version of this article includes the following figure supplement(s) for figure 3:

**Figure supplement 1.** A linear model predicts variant fitness.

non-cognate partner. As a first pass to determine whether the wild-type variant is optimal for specificity, we calculated a specificity gap ($W_{ParE3} – W_{ParE2}$) for every variant in the library. This analysis revealed a small fraction (2.6%) of the library (*Figure 3D*) with a wider specificity gap than the wild-type antitoxin. However, given the nature of our assay, it may be difficult to reliably compare highly fit variants, given that all variants that fully neutralize toxin are likely to have maximal apparent fitness. Thus, the measured fitness of the wild-type variant could be an underestimate of its ability to antagonize the cognate toxin. To probe this possibility, we built a linear model of variant fitness against the cognate toxin, using k-fold cross-validation with five folds (*Figure 3—figure supplement 1A*). Although a purely additive model of residue contributions was highly predictive of variant fitness ($R^2 = 0.89$, SD between folds ±0.003; *Figure 3—figure supplement 1B*), the model was weakest for the fittest variants, likely due to diminishing returns for highly favorable residues, which may reflect a limitation of our assay or a non-linear relationship between the free energy of binding and the fitness values measured. We observed a similar effect in a linear model for variant fitness against ParE2 (*Figure 3—figure supplement 1C–D*).

The success of the linear model in predicting variant fitness implies that individual substitutions are largely independent, that is, epistasis in our library is relatively limited. To better quantify epistasis, we plotted the difference between the fitness of each variant predicted assuming independence and the observed fitness of that variant. We considered all possible double and triple mutants relative to the wild type (*Figure 3—figure supplement 1E–F*). In each case, the distribution of the differences calculated was tightly centered around 0, further suggesting that, on average, there is minimal epistasis or interdependency of mutations. This finding contrasts with the epistasis observed in other proteins (*Olson et al., 2014*; *Podgornaia and Laub, 2015*; *Salinas and Ranganathan, 2018*), and may reflect the fact that each position varied in our ParD3 library interacts with a non-overlapping set of residues in ParE (*Figure 1*; also see Figure 5 below).

## Leveraging a distribution of fitness effects to measure contributions to specificity

We sought a method to better capture the impact of each interface residue on specificity. Introducing individual mutations offers limited information about the role of a residue in determining specificity, particularly due to the intrinsic degeneracy of the interface and the inability to distinguish between high fitness variants. For example, given that ParD3 D61 forms a salt bridge with ParE3 R58 in the native ParD3-ParE3 interaction (*Figure 2A*), mutations at position 61 (library position 1) should weaken the cognate interaction. However, ParD3 D61I neutralizes wild-type ParE3 in vivo (*Figures 1E* and *2E*) to a degree indistinguishable from wild-type ParD3 in our competition experiment, likely reflecting the robustness of the cognate interface in the context of our assay. A similar effect could, in principle, impact the assessment of how individual mutations affect non-cognate interactions.

To overcome these limitations, we leveraged the inherent redundancy in our saturation mutagenesis library, namely that each substitution occurs in the context of ~400 unique backgrounds. For example, the D61I mutation occurs in our library with all possible combinations of residues at positions 64 and 80 (*Figure 4A*). We calculated the value $\Delta W$ for each substitution in each possible background, for example, for D61I in the context where positions 64 and 80 are alanine, $\Delta W = W_{DAA} - W_{IAA}$. To understand the role of each antitoxin residue, both in terms of mediating cognate toxin binding and preventing non-cognate toxin binding, we can measure and assess the distribution of fitness effects ($\Delta W$ values) for mutating each residue in each possible background. For this and all further analyses, we do not consider variants containing prolines because all three library positions reside in α-helices, and prolines tend to break or kink helices; these substitutions could trivially affect antitoxin stability rather than molecular recognition. Thus, we ultimately considered ~361 library contexts per substitution.

We first used this approach to examine the fitness effects of substituting each ParD3 residue with the corresponding residue in ParD2, that is, D61I, K64L, and E80K substitutions. If an antitoxin residue generally serves as a negative specificity element, we predict that a substitution in that residue will have an overall positive distribution of fitness effects when the library is queried against the non-cognate ParE2 (*Figure 1A*). If an antitoxin residue generally serves as a positive element, we predict that its substitution will produce an overall negative distribution of effects when the library is queried against ParE3. Residues could, in principle, serve as both positive and negative elements. To summarize the fitness effects of a given substitution in the 361 different possible contexts, we fit each distribution to a skew normal distribution and assessed the mode as a measure of central tendency (*Figure 4B*; *Figure 4—figure supplements 1–2*).

We found that all three library positions serve as positive elements that promote the cognate interaction (*Figure 4B*). Swapping the ParD3 residue with the ParD2 residue had a typical fitness cost, or produce a change in the mode, of −0.63, −0.71, and −0.39 at positions 1, 2, and 3, respectively, when the library was tested against ParE3. These distributions were almost universally negative, and although there was substantial overall variability, each had relatively narrow interquartile ranges (0.17–0.25). Thus, swapping each ParD3 specificity residue for the corresponding residue in ParD2 has a generally detrimental effect on fitness. Importantly, however, each of these substitutions in the wild-type background (i.e., DKE → IKE, DLE, and DKK) was approximately 0 (*Figure 4B*), demonstrating the importance of using the entire saturation mutagenesis library to characterize the nature of a given substitution.

Although all three positions promote interaction with the cognate partner, only two serve as negative elements that help prevent interaction with the non-cognate toxin ParE2. At position 1, swapping the ParD3 residue with the corresponding ParD2 residue (D61I) produced a distribution of fitness changes with a mode of −0.01 (statistically indistinguishable from 0) with an extremely narrow interquartile range of 0.11 (*Figure 4B*, left). By contrast, the K64L and E80K mutations had distributions with modes of 0.57 and 0.50, respectively (*Figure 4B*, middle and right). Taken together, these analyses indicate that position 1 functions primarily as a positive element, whereas positions 2 and 3 have both positive and negative roles in ParD-ParE specificity.

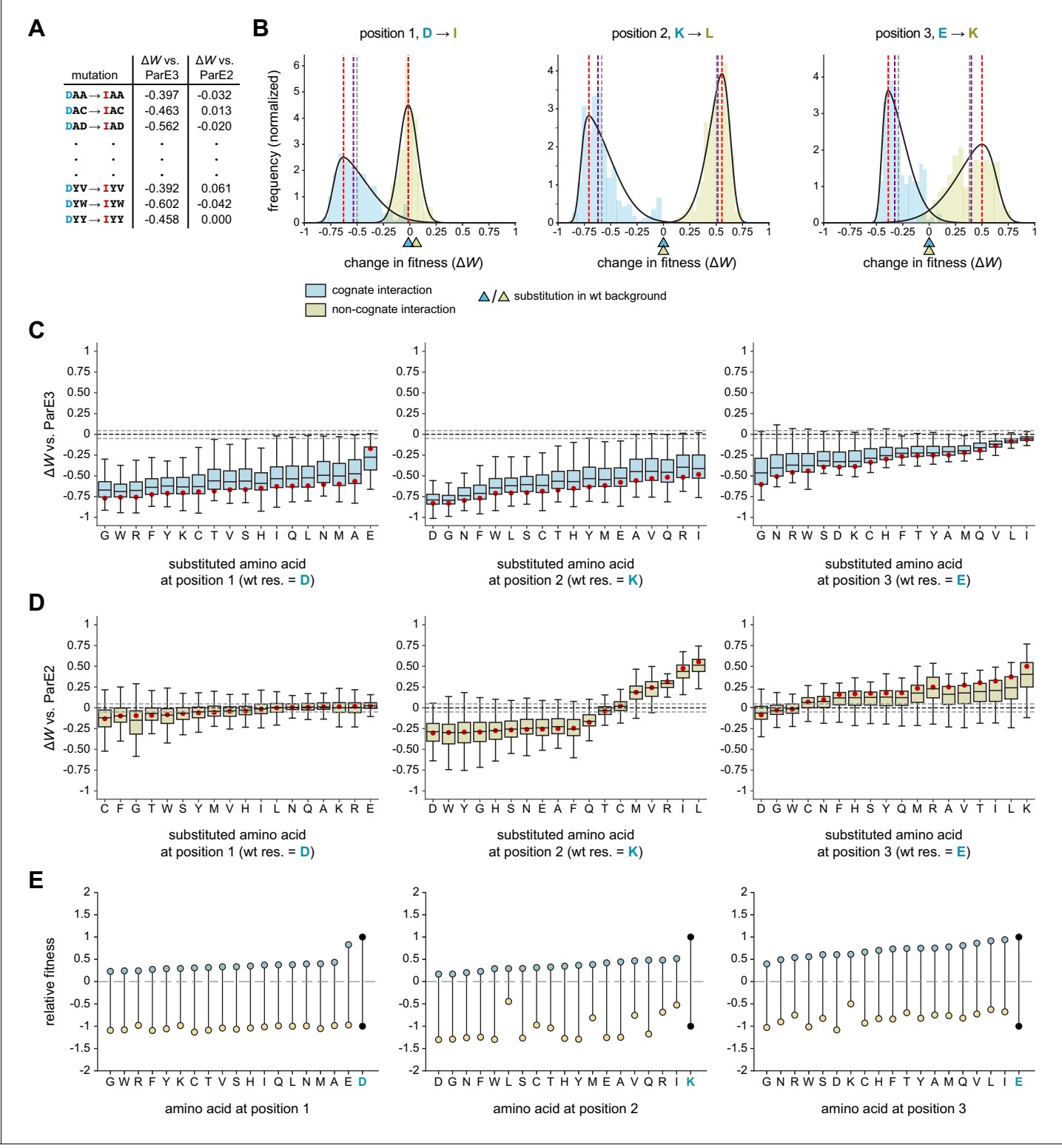

**Figure 4.** Systematic dissection of interface substitutions. (A) Example illustrating how each substitution occurs in hundreds of contexts in the saturation mutagenesis library. (B) Distribution of fitness effects for the substitutions indicated (each replacing a specificity residue in ParD3 with the corresponding residue in ParD2) in all possible contexts, fitted to skew normal distributions. The effect on the cognate interaction is shown in blue, and the effect on the non-cognate interaction is shown in gold. Mean (gray), median (purple), and mode (red) are indicated. Triangles denote the effect of each substitution in the wild-type background (DKE). (C–D) Box plots showing the distribution of fitness effects for each substitution. For each substitution, the effects on fitness against ParE3 (C, blue) and ParE2 (D, gold) are shown. The mode of the fitted skew normal distribution is indicated in

*Figure 4 continued on next page*

*Figure 4 continued*

red. Dashed lines represent the threshold for a mode not expected by chance (<5% FDR based on 1000 permutation tests). Outliers (1.5 * IQR) are hidden for clarity but were included in all quantitative analyses. (E) Diagram illustrating how each substitution affects fitness for both ParE3 and ParE2. Starting fitness of the wild-type antitoxin is arbitrarily set to $W = 1$ for ParE3 and $W = -1$ for ParE2 (black dots). Blue and gold points represent shifts in fitness for variants when tested against ParE3 and ParE2, respectively, based on the mode effect size measured in (C) and (D).

The online version of this article includes the following figure supplement(s) for figure 4:

**Figure supplement 1.** Characterizing the distribution of fitness effects for every amino acid substitution in the ParD3 library.

**Figure supplement 2.** Comparison of linear model coefficients to parameters estimated from the distribution of fitness effects.

**Figure supplement 3.** Library estimates of residue fitness and optimality.

## Systematically estimating the effects of all interface substitutions on specificity

The analyses just presented only considered substituting a residue in ParD3 with the corresponding residue in ParD2. Next, we wondered whether the effects documented (*Figure 4B*) were idiosyncratic or generalizable to all substitutions. In other words, do all amino acid substitutions at position 1 (D61) hinder the cognate interaction, and do all substitutions at positions 2 and 3 (K64 or E80) have reciprocal effects on the cognate and non-cognate interactions? Generally, is each ParD3 interface position 'optimized' for specificity? To answer these questions, we measured the distribution of fitness effects, again by considering all ~361 contexts in the library, for each amino acid substitution at each library position (*Figure 4C–D*).

We observed distinct patterns of effects at each library position. At position 1, we found that mutating 61D to any residue hindered ParD3-ParE3 binding (*Figure 4C*). Substituting glutamate for aspartate at this position was the least unfavorable mutation when considering the entire distribution of effects (*Figure 4C*), as expected given that ParD3 61D likely forms a salt bridge with ParE3 58R (*Figures 1D* and *2A*). With respect to the non-cognate interaction, every distribution had a mode close to or indistinguishable from 0 (*Figure 4D*). In no case did a substitution tend to improve binding to ParE2. However, eight substitutions (to C, F, G, T, W, S, Y, or M) each tended to further destabilize the ParD3-ParE2 interaction. Interestingly, substituting ParD3 position 1 with isoleucine, the corresponding residue at this position in ParD2, did not confer a noticeable fitness advantage relative to other substitutions in terms of promoting an interaction with ParE2. More generally, these results suggest that the ParD3 residue 61D is likely not important for discriminating against the non-cognate toxin. We conclude that position 1 in the antitoxin primarily supports the cognate interaction with ParE3.

At position 2, the distributions of fitness effects for all substitutions were, as for position 1, nearly universally negative in terms of interacting with the cognate ParE3 (*Figure 4C*). With respect to interacting with the non-cognate toxin ParE2 however, there were striking differences among the different substitutions (*Figure 4D*). The substitutions from lysine to five different residues (L, I, R, V, M) each significantly improved ParD3 binding to ParE2. For the substitutions of K with L, I, or R, the entire distributions were ≥0 indicating that these substitutions promoted an interaction with ParE2 in all ~361 contexts. By contrast, substitutions to 11 amino acids generally decreased the fitness of ParD3 variants versus ParE2. These findings imply that K64 is a negative design element, but that lysine is not the optimal residue for discriminating against ParE2 (discussed further below). Notably, performing this analysis using the 130 genetic backgrounds found in our previous library (*Aakre et al., 2015*) did not reliably capture the effects of each substitution (*Figure 4—figure supplement 3*).

Finally, for position 3, the distributions of fitness effects with respect to the cognate interaction were again generally negative, as with positions 1 and 2 (*Figure 4C*). However, the typical values and the lowest values of the distributions were notably less severe for position 3, indicating that substitutions at this position are generally more forgiving with respect to maintaining the interaction with the cognate toxin. Conversely, nearly all substitutions at position 3, in virtually all contexts, led to increased interactions between ParD3 and ParE2, consistent with this position in ParD3 helping to enforce specificity by preventing an interaction with ParE2.

Taken all together, our results and analyses provide a comprehensive map of how substitutions impact ParD-ParE interaction specificity and demonstrate that the three positions each play

a distinct role in enforcing this specificity. To summarize these results, and to explore how each substitution simultaneously affects the cognate and non-cognate interactions, we plotted the typical (i. e., mode) effects of each substitution on ParD3 fitness versus. ParE3 and ParE2 (*Figure 4E*). We arbitrarily set the starting values of the wild-type variant DKE to 1 for ParE3 and −1 for ParE2. Increasing fitness versus the cognate toxin or decreasing fitness versus the non-cognate toxin would each increase overall specificity (*Figure 1A*). Notably, the wild-type residue at each position is optimal for ParE3 binding, such that no substitutions—considering the distribution of fitness effects—improve the fitness of ParD3 queried against this toxin. We do find many substitutions in ParD3 that can further destabilize the non-cognate interaction. However, every such substitution has a substantial cost in fitness when queried against the cognate ParE3. Thus, we conclude that the wild-type ParD3 may provide optimal specificity with respect to ParE3 and ParE2. One limitation of this approach is that each type of substitution is compared to the wild-type amino acid only in its wild-type context (e.g., AXX mutations are compared to DKE), which may overestimate the favorability of the wild-type residue. However, performing a similar analysis in which fitness distributions for each amino acid at each position were compared to the fitness distribution for the wild-type residue (e.g. AXX compared to DXX) produces similar results and conclusions (*Figure 4—figure supplement 3E–G*).

## Structural basis of positive and negative interaction elements

Our saturation mutagenesis results produced several hypotheses about the contributions of individual residues and residue combinations to ParD-ParE interaction specificity. To probe these hypotheses further and to understand how positive and negative elements of specificity function at a structural level, we sought to compare the packing of interface residues in co-crystal structures of ParD3-ParE3 and ParD2-ParE2. For simplicity in comparisons, we use residue position numbers based on the alignment in *Figure 1C* for all analyses.

In the ParD3-ParE3 co-crystal structure, which we solved previously (*Aakre et al., 2015*), library position 1 (D61) and position 2 (K64) both reside in α-helix-2 of the antitoxin in a hydrophobic side chain pocket (*Figure 5A*). D61 forms two salt bridges with toxin R59, while K64 forms salt bridges with toxin E75 and/or E89. These inter-chain salt bridges flank the antitoxin residue W60, permitting extensive hydrophobic contacts between the large aromatic residue and the aliphatic side chains. Toxin residues A62, L73, and V91 also form hydrophobic contacts with the antitoxin residue W60. Previous work has shown that the burial of tryptophan residues is highly energetically favorable and helps drive protein-protein interactions (*Cunningham and Wells, 1989*). Thus, we posit that D61

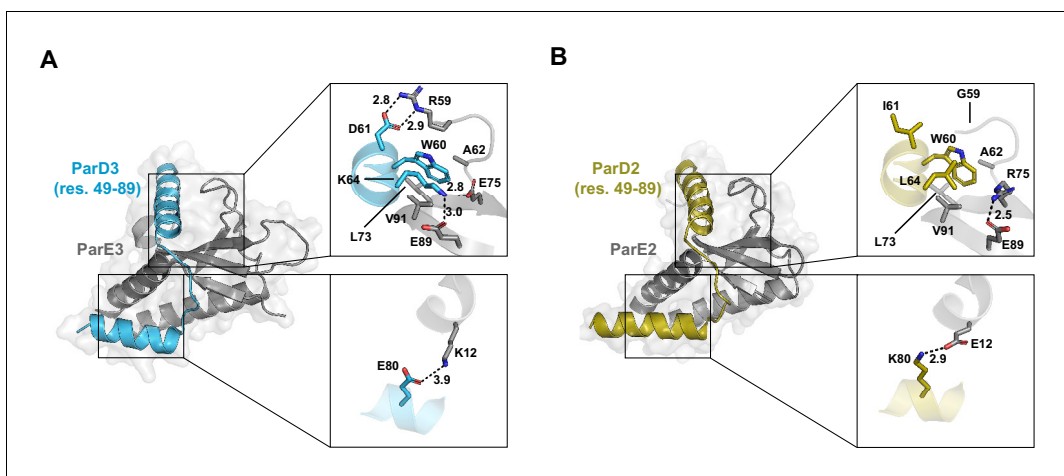

**Figure 5.** Comparing the interfaces of paralogous toxin-antitoxin complexes. (**A**) The interface between ParD3 and ParE3 (PDB: 5CEG). Top inset, library positions 1 and 2, with neighboring residues. Bottom inset, library position 3. Salt bridges are indicated by dashed lines. (**B**) Same as in (**A**), illustrating the corresponding regions of ParD2-ParE2 (PDB: 6X0A).

The online version of this article includes the following figure supplement(s) for figure 5:

**Figure supplement 1.** Crystal structures of the ParD2-ParE2 interface.

and K64 serve as positive specificity elements by both forming direct inter-protein contacts with the toxin and decreasing solvent accessibility of W60. Substitutions at positions 1 and 2 were nearly uniformly (i.e., in all ~361 contexts) detrimental to interacting with ParE3 likely because they disrupt the salt bridges normally formed by D61 and K64, perturb the hydrophobic pocket surrounding W60, or both. The third library position, E80, resides on the antitoxin α-helix-3 and forms a salt bridge with toxin residue K12. Mutations at this position are generally less detrimental than mutations at the other positions, potentially due to the absence of an equivalent hydrophobic binding pocket.

To compare the binding interfaces formed by the paralogous interacting pairs, we initially solved a 2.4 Å co-crystal structure of ParD2-ParE2 (*Figure 5—figure supplement 1A*; also see *Source data 5–6*). The toxins and the C-terminal half of the antitoxins, which contains all toxin-binding residues, were remarkably similar between the two cognate complexes (rmsd = 0.812 Å). However, we were unable to resolve ParD2 residues 26–42 (*Figure 5—figure supplement 1A–C*), likely because this region does not have any protein-protein contacts and adopts multiple conformations. We speculate that ParD2 residue 47P (which is 47E in the ParD3 structure) is helix-breaking and contributes to the instability of this region. Although our model for the toxin and toxin-binding region of the antitoxin (residues 43–89) were an excellent fit to the electron density (*Figure 5—figure supplement 1C*), noisy electron density for the N-terminal portion of the antitoxin prevented adequate refinement of the entire structure (see Materials and methods). To produce a second model for the ParD2-ParE2 interface, we initially pursued truncated versions of ParD2 in complex with ParE2 but could not obtain crystals. Ultimately, we produced a second co-crystal structure to 2.9 Å resolution of ParE2 in complex with a ParD2 variant, ParD2*, modified to have the N-terminal 48 residues of ParD3 (PDB: 6X0A; *Figure 5—figure supplement 1D*; Table S1). In the original ParD3-ParE3 structure, this region of ParD3 is crystallographically well-behaved and forms extensive contacts with a second antitoxin chain. Importantly, both structural and coevolutionary data suggest that there are no contacts between ParE2 and residues 1–45 of the antitoxin, and the modified structure was indistinguishable from the original structure in the overlapping regions (*Figure 5—figure supplement 1E–F*).

The ParD2*-ParE2 structure reveals a similar hydrophobic binding pocket to ParD3-ParE3 surrounding W60 and involving toxin residues A62, L73, and V91 (*Figure 5B*). However, the interactions corresponding to the library positions are markedly different between the paralogous interacting pairs. ParD2 residue 61 (corresponding to position 1 in our library) is isoleucine, versus aspartate in ParD3. Unlike ParD3 D61, which forms a salt bridge with R59 in ParE3, ParD2 I61 does not form a salt bridge. Substitutions of ParD3 residue 61 do not improve binding to ParE2, likely because ParE2 G59 provides flexibility to this region and space to accommodate almost any residue. ParD2 residue 64, which corresponds to the second library position, is leucine, and interacts hydrophobically with W60 (distance = 4.2 Å). Unlike ParD3 K64, which interacts with two glutamate residues, ParD2 L64 does not contact the toxin. Instead, ParE2 E89 forms an intra-chain salt bridge with R75. The hydrophobic interactions between L64 and W60 in the ParD2*-ParE2 structure may explain why ParD3 K64 mutations to the hydrophobic residues leucine, isoleucine, and methionine generally improve ParD3-ParE2 binding (*Figure 4D*). Intriguingly, arginine mutations at this position also improved ParD3-ParE2 binding. We speculate that, due to its longer side chain, arginine may be more effectively neutralized than lysine by ParE2 E89. On α-helix-3 in ParD2*-ParE2, K80 forms a salt bridge with E12, analogous to the salt bridge formed by ParD3-ParE3 with E80-K12 (*Figure 5A*). Importantly, the charges of the residues are flipped between the complexes, providing a simple mechanism for how ParD3 excludes ParE2 at this position.

## Discussion

### A combinatorially complete library reveals substitution effects across hundreds of contexts

Paralogous interacting proteins face the challenge of maintaining the desired interaction while avoiding unwanted cross-talk. This specificity is often accomplished via molecular recognition using a discrete set of amino acids at the protein-protein interface. These specificity-determining residues must inherently stabilize the cognate interaction (positive elements), destabilize the non-cognate interaction (negative elements), or both. Determining which of these roles individual specificity residues play is a difficult problem that can be confounded by epistasis and the degeneracy of the

interface, which can mask both beneficial and detrimental effects of individual substitutions. Prior work has shown that examining a substitution across multiple genetic contexts can improve estimates of its effects on an interface. For instance, one study found that including additional mutational data (using a library of 2–6 amino acids at four different interface positions) improved the performance of a position weight matrix to predict binding of a BH3 peptide to either of two Bcl-2 family members (*Dutta et al., 2010*). We expanded on this approach by screening a combinatorially complete library of ~$20^3$ interface variants using ParD-ParE toxin-antitoxin systems as a model. In this library, each substitution occurs in ~$20^2$ different backgrounds, which led to relatively narrowly-distributed, unimodal distributions of fitness effects in each case (*Figure 4C–D*). Based on these distributions, we determined the typical effect of mutating a given specificity residue to every other amino acid. Notably, most substitutions in the wild-type background had little or no effect on either the cognate or non-cognate interaction, which underscores the value of our approach for elucidating the contribution of individual substitutions to specificity (*Figure 2E and H*; *Figure 4*).

For the ParD3 antitoxin, we found that all three library positions are positive elements, as all substitutions at each position tended to destabilize interaction with the cognate toxin, ParE3 (*Figure 4C*). Additionally, each wild-type residue appears optimal for this role, suggesting that there is normally a high fitness cost of inadequately neutralized toxin. With regard to the non-cognate interaction, we found that positions 2 and 3—but not position 1—serve as negative elements in determining specificity (*Figure 4D*). In other words, the wild-type residues at positions 2 and 3 in the ParD3 antitoxin help inhibit an interaction with ParE2, with many (position 2) or nearly all (position 3) substitutions increasing the interaction of ParD3 with ParE2. These results argue against the notion suggested previously for other paralogous interacting proteins (*Hochberg et al., 2018*; *Sergeeva et al., 2020*) that it is difficult to introduce mutations in one protein that improves binding to the cognate partner without also improving binding to the non-cognate partner.

Examining comprehensive, combinatorial libraries also enabled us to ask whether specificity is optimal, at least with respect to the three positions randomized in our library. Overall, ParD3 specificity appears to be optimized with respect to antagonizing the cognate toxin ParE3 and avoiding the non-cognate toxin ParE2 (*Figure 4E*). When considering the distribution of fitness effects, no substitutions further stabilized the cognate interaction. Several substitutions improved discrimination against the non-cognate ParE2 (e.g., substitutions to D, W, and Y at position 2 – see *Figure 4D*), but each substitution led to diminished interaction with the cognate toxin. Whether ParD3 is globally optimized for specificity with respect to all other homologous ParE toxins—for instance, the more distantly related ParE1—remains to be tested. Specificity may not be globally optimized, as was shown recently for paralogous bacterial two-component signaling proteins (*McClune et al., 2019*).

Importantly, our assessment of an interface affinity and discrimination is derived from an in vivo fitness assay, as opposed to an in vitro binding assay. This was partly a necessity, as it is technically difficult to purify large quantities of many toxins given their inherent toxicity, and to purify antitoxins, which are often unfolded in the absence of their cognate toxins (*Cherny and Gazit, 2004*; *De Jonge et al., 2009*). Binding assays have been reported for some TA systems (e.g., *Brown et al., 2013*), but this approach is not scalable to assaying many antitoxin variants in parallel, as done here. Further, even precise quantitative measurements of binding affinity may not predict whether two systems will be insulated in vivo because of non-linear relationships between affinity in vitro and fitness in vivo. In this regard, the use of an in vivo assay is advantageous relative to in vitro affinity measurements as it may better reflect what matters to a cell. However, our in vivo fitness assays are unlikely to be as sensitive as what occurs over long time-scales in nature with large effective population sizes.

We also note that our in vivo assays rely on the induced expression of toxins and antitoxins in a heterologous host, and so may not accurately reflect their concentrations in their native cellular contexts. Nevertheless, the relative behavior of ParD3 variants measured here is likely to apply in whatever expression context they arise (*Figure 2—figure supplement 2*). Finally, we note that in principle some mutations could affect protein abundance rather than toxin affinity, although the positions mutated are all solvent-exposed residues that are unlikely to affect protein folding and stability.

## Structural basis of protein interaction specificity

Our results demonstrate that each position examined in ParD3 made qualitatively different contributions to specificity. To better understand the patterns documented in our library approach and to develop a structural basis for them, we produced a co-crystal structure of the ParD2-ParE2 complex and compared it to the ParD3-ParE3 structure we produced previously (*Aakre et al., 2015*). Although these systems do not cross-talk in vivo (*Figure 1E*), the complexes share highly similar secondary and tertiary structures. This overall structural similarity underscores the importance of just a few interface residues in dictating interaction specificity, as has been observed for many other systems (e.g., *Brasch et al., 2018*; *Bridgham et al., 2006*; *Cosmanescu et al., 2018*).

The first position, ParD3 residue D61, is optimal for cognate toxin binding but does not help discriminate against the non-cognate toxin. In the ParD3-ParE3 structure, ParD3 D61 forms a salt bridge with R59 in ParE3. However, in the ParD2-ParE2 structure, there is no connection between the corresponding two residues. In fact, the arginine in ParE3 is replaced by a glycine in ParE2, G59, which likely confers local flexibility and tolerance for diverse amino acids at position 61 in the antitoxin.

At the second position, ParD3 residue K64 is both a positive and negative element with respect to specificity. This lysine is optimal for the cognate interaction but not for non-cognate discrimination. Some mutations would better discriminate against the non-cognate interaction (*Figure 4D*), although each would come at the expense of cognate binding (*Figure 4C*). In the ParD3-ParE3 structure, ParD3 K64 forms an inter-chain salt bridge with E75 and E89 in the toxin; the aliphatic portion of the lysine side-chain also contributes to the hydrophobic packing of a conserved tryptophan, W60. Because ParD2 encodes a leucine at position 64, there is no inter-chain salt bridge; instead, there is an intra-chain salt bridge between toxin residues R75 and E89. This provides a structural explanation for why hydrophobic residues (e.g., L, I, V, and M) at ParD3 position 64, which are less likely to interfere with this salt bridge, are more compatible with binding to ParE2. Positively-charged residues (R, K) at position 64 in the antitoxin are also well-tolerated, with respect to binding ParE2, as they may be neutralized by E89 in the toxin.

Finally, the ParD3 residue E80, which corresponds to the third library position, offers maximal interaction with the cognate ParE3 and near-maximal discrimination against ParE2. The only residue which offered significantly more discrimination was aspartate, which shares its negative charge. ParD3 residue E80 forms a salt bridge with K12 in ParE3, while ParD2 K80 forms a salt bridge with E10 in ParE2. In other words, both complexes form salt bridges, but with reversed charges. Opposite salt bridges provide a logical strategy for two paralogous complexes to simultaneously maximize both affinity and discrimination.

Our combined genetic and structural data allow us to predict how ParD3 discriminates against other paralogous toxins. We hypothesize that ParD3 maintains insulation from ParE1 using a different subset of interfacial residues than it uses for ParE2. For instance, both ParD2-ParE2 and ParD3-ParE3 encode a tryptophan at antitoxin position 60 and a leucine at toxin position 73, which lie in a hydrophobic pocket with similar architecture in both cognate complexes (*Figure 5A–B*). However, ParD1-ParE1 encodes isoleucine at antitoxin position 60 and phenylalanine at toxin position 73 (*Figure 1C*). Thus, we speculate that the ParD1-ParE1 complex also forms a hydrophobic pocket but with a fundamentally different architecture than the ParD2-ParE2 and ParD3-ParE3 complexes. Steric clash between the residues of this hydrophobic pocket could serve as a common mechanism of insulating ParD3-ParE3 and ParD2-ParE2 systems from ParD1-ParE1. The use of different subsets of residues to discriminate against different partners has been observed in other systems (*Sergeeva et al., 2020*; *Zhang and Palzkill, 2004*).

## Concluding remarks

Our work demonstrates the novel application of a high-throughput in vivo approach to determining the contributions of individual residues to interaction specificity. Understanding specificity in natural proteins can inform the design of proteins and synthetic signaling pathways. Our results reinforce the notion that negative interaction elements are necessary to maintain insulation between two paralogous toxin-antitoxin systems. Previous work similarly found that explicit negative design is required to prevent undesired interactions between paralogous proteins (*Grigoryan et al., 2009*). Elucidating the basis of interaction specificity can also provide insight into how paralogous proteins

have become insulated during evolution. For toxin-antitoxin systems, as with many interacting proteins, new systems emerge through duplication and divergence, with the paralogous pairs becoming insulated through the forces of drift and selection. Mapping the basis of specificity, as done here, will enable efforts to reconstruct the phylogenetic history of such duplication-divergence events and to elucidate how insulation proceeds through a series of stepwise mutations.

## Materials and methods

### ParD-ParE coevolution and phylogeny

ParD-ParE specificity residues, those residues showing strongest coevolution between proteins, were identified using GREMLIN (http://gremlin.bakerlab.org) using *Mesorhizobium opportunistum* ParD3 and ParE3 as input. We performed eight iterations with an E-value cutoff of $1 \times 10^{-20}$ and isolated residue pairings with a probability score >0.95. To construct a ParE protein tree, we first used HMMER (http://hmmer.org) to identify and align all α-proteobacteria homologs of *M. opportunistum* ParE2 in the ProGenomes Database. We then built a protein tree from all homologs using a trimmed alignment (positions represented in <50% of sequences were removed) in FastTree 2 (*Price et al., 2010*). The protein tree allowed us to categorize the resulting homologs as orthologs of ParE2, ParE3, or neither based on membership in either the ParE2 or ParE3 sister taxa. A species tree was generated for a concatenated alignment of 15 highly conserved single-copy bacterial genes (*atpD, dnaA, rpsK, rpsD, ruvA, leuS, yeaZ, rpsF, rplI, radA, tsf, pyrH, yhhF, coaD, frr*). HMMER was used to identify and align orthologs of these genes in α-proteobacteria species in the ProGenomes Database. The concatenated alignment was manually trimmed to remove positions represented in <50% of sequences and positions with <25% conservation, and a tree was generated using FastTree two for relevant species.

### Bacterial strains and media

*Escherichia coli* strains were grown in an M9L medium (1× M9 salts, 2 mM MgSO$_4$, 100 uM CaCl$_2$, 10% v/v LB) supplemented with 0.1% casamino acids and 0.4% glycerol at 37°C, unless otherwise indicated. ParE toxins were cloned between SacI and HindIII sites in the arabinose-inducible pBAD33 vector, and ParD antitoxins were cloned between SacI and HindIII sites in the IPTG-inducible pEXT20 vector. Plasmids were co-transformed into *E. coli* TOP10 cells and maintained with chloramphenicol (30 μg/mL) and carbenicillin (100 μg/mL). Single colonies were grown overnight at 30°C in an M9L medium supplemented with 0.4% glucose and antibiotics. The following day, cultures were washed in media without glucose, serially diluted, and spotted onto M9L plates supplemented with antibiotics and 0.4% glucose (toxin repression), 0.2% arabinose (toxin induction), or 0.2% arabinose and 100 μM IPTG (toxin and antitoxin induction). After 1 day, toxin-antitoxin interactions produced robust growth on the M9L plates with 0.2% arabinose and 100 μM IPTG. We observed no intermediate growth phenotypes for strains in (*Figure 1E*).

### Library construction

Residues targeted in the randomized library were selected using GREMLIN, followed by identification of specificity residues that differ between ParD3 and ParD2. The ParD3 saturation mutagenesis library was assembled using two rounds of PCR with the KAPA HiFi HotStart ReadyMix PCR Kit (Roche). The first round of PCR replaced the targeted residues with NNK codons and produced a linear fragment with 5′ and 3′ BtgI sites. To decrease the frequency of wild-type ParD3 in the pooled library, we used a mutated, non-functional gene (ML3304) with a deletion between codons 64–80 and frameshifted between residues 80–93 as the template. The fragment was then re-circularized via digestion with BtgI followed by ligation with T4 DNA ligase. To ensure strand complementarity, we performed a second round of PCR amplifying the entire ParD3 gene. The amplicon was then subcloned into pEXT20 between the SacI and HindIII sites and transformed into One Shot TOP10 Electrocomp *E. coli* (Invitrogen). Cultures were recovered overnight at 30°C in LB containing 0.4% glucose and carbenicillin to produce ML3310. The yield was ~$10^6$ total colonies (theoretical library size = $8 \times 10^3$ at the codon level, $3.3 \times 10^4$ at the nucleotide level). Cells harboring the plasmid library were then electroporated with a dialyzed plasmid containing an arabinose-inducible toxin.

The transformations were recovered 1 hr in SOC media, followed by overnight growth at 30°C in LB supplemented with 0.4% glucose and antibiotics.

## Library selection and analysis

To assess the ability of each library variant to antagonize a given ParE toxin, we performed competitive growth assays in liquid cultures. To this end, 50 mL of an overnight culture of the library was washed in 50 mL M9L without glucose and diluted to an $OD_{600}$ of 0.03 in 200 mL M9L supplemented with antibiotics. To induce antitoxin expression, 100 µM IPTG was added and cells were grown at 37°C with shaking for 90 min. Toxin expression was then induced with 0.2% arabinose. Optical density was monitored every 30 min between t = 0–5 hr and every 60 min between t = 5–10 hr. To prevent cells from entering the stationary phase, cells were diluted 1:10 in fresh media 150 min after induction. To identify changes in variant frequency over the course of the experiment, plasmids were extracted (Zymogen) from 50 mL samples collected at t = 0 and t = 10 hr. The entire ParD3 gene was used as a template for PCR (12 cycles) with custom barcoded primers containing Illumina flow-cell adaptor sequences. To increase the diversity of the amplicon pool, the PCR also incorporated 4–6 random nucleotides to the 5' end of each ParD3 gene. Samples were then pooled and sequenced on a single lane of a NextSeq flow cell.

Multiplexed reads were sorted based on an exact match to a six-letter barcode. Reads were then filtered to remove sequences that (a) do not encode an 'NNK' codon in any of the three targeted positions, or that (b) contain any non-synonymous mutation in the remaining 91 codons. The frequency of each variant in each sample was then counted and normalized by the total number of reads per sample. Next, we assigned a raw fitness score to each variant as described previously (*Aakre et al., 2015*; *van Opijnen et al., 2009*) and transformed those fitness values such that the median *W* value for variants encoding a stop codon = 0, and the *W* value for the wild-type or predicted best sequence = 1 (DKE for ParE3, ILK for ParE2).

Linear models were built using k-fold cross-validation with five folds, which allows each variant to serve as part of the training and test sets, and permits error estimation for the model coefficients. Models assumed additive relationships between residues. To summarize the fitness effects of substitutions, we fit each distribution to a skew normal distribution, which generalizes the normal distribution to allow for non-zero skewness. We posit that this probability distribution best summarizes the data because our distributions of mutation effects skew toward 0 change in fitness, likely due to the inherent sensitivity limits of the assay for changes at either end of the fitness distribution. Indeed, we find that variants most resistant to change in fitness tend to have the highest and lowest fitness scores (*Figure 4—figure supplement 1A–B*). We report the mode of the skew normal distribution as our measure of central tendency for the effect of each substitution, but note that it correlates extremely well with other measures such as mean and median (*Figure 4—figure supplement 1C–D, F–G*) as well as the coefficients from our linear model (*Figure 4—figure supplement 2*). To determine whether the modes for each distribution deviate significantly from 0, we generated a null distribution by shuffling the variant names in our data and again measuring the modes of fitness distributions. We call an effect significant if its mode lies below the 5% or above the 95% percentile of this null distribution.

## Crystallization of ParD2-ParE2 and chimeric ParD2*-ParE2

To purify the native ParD2-ParE2 complex, the genes were cloned co-operonically under a single IPTG-inducible promoter in the pETDuet vector between the NcoI and HindIII sites (ML3297). Protein expression was induced in T7 Express cells (NEB) with 0.2 mM IPTG at 16°C. Cells were harvested 16–24 hr after induction. Frozen cell pellets were resuspended in Tris-buffered saline (50 mM Tris-HCl [pH 7.6], 150 mM NaCl) with 1 mM PMSF, and lysed via three passages through an LM20 Microfluidizer (Analytik). The precipitate was pelleted at 30,000 g for >1 hr and cleared lysate was purified on a $Ni^{2+}$-NTA column followed by gel filtration on a Superdex 200 10/300 (GE Healthcare Life Sciences) equilibrated with storage buffer (10 mM Tris-HCl pH 8.0, 150 mM NaCl, 200 mM imidazole). Fractions containing ParD2-ParE2 were identified via SDS-PAGE.

Initial sitting-drop, vapor-diffusion crystallization screens were set up with a Phoenix drop-setting robot (Art Robbins Instruments). Hits were optimized in hanging-drops. For screening, we mixed 0.15 µL of protein (7.5 mg/mL) with 0.15 µL reservoir solution from commercial crystallization kits.

Crystals appeared within ~3 d in condition 77 of the Protein Complex Suite (Qiagen) and ~12 hr in condition 59 of the PegRx HT kit (Hampton). Optimized crystals were obtained in 2 µL drops (1 µL protein + 1 µL well solution) at 18°C in two conditions. Crystals for Br/SAD data collection were grown using 0.1 M HEPES (pH 7.40), 0.85 M KCl, 1 M AmSO₄, 0.35 M KBr as the well solution. Native crystals were grown using 1.93 M AmSO₄, 0.1 M BIS-TRIS pH 6.50, and 2–3% w/v PEG MME 550. Crystals were cryo-protected in well solution variants substituting 1.5 M LiSO₄ (Br/SAD) or 2.0 M LiSO₄ (Native) for AmSO₄.

X-ray diffraction data were collected on the 24-ID-C beamline at APS (NE-CAT). A 2.8 Å Br/SAD data set was collected at a wavelength of 0.920 Å, and a native data set was collected to 2.4 Å resolution at 0.979 Å. Data were indexed, integrated, and scaled using XDS (*Kabsch, 2010*). The two data sets belong to the same space group (I4₁22) but have slightly different unit cells, making phasing by SIRAS not feasible. The PHENIX tool XTriage estimated the anomalous signal in the SAD data extended to 4.3 Å (*Adams et al., 2010*). The AutoSol function of PHENIX was used to locate bromide sites, calculate SAD phases, and refine them with density modification (*Terwilliger et al., 2009*). The resulting density modified map was sufficiently clear to identify a bundle of three alpha-helices which were built in Coot (*Emsley and Cowtan, 2004*) as poly-alanine and input back into AutoSol for phase recombination with the SAD phases. In subsequent rounds of phase recombination and density modification, a nearly complete model of the complex was built and refined in PHENIX. Throughout this process, the structure of the ParD3-ParE3 complex (PDB: 5CEG) served as a guide to identify noisy features. The resulting model was used to phase the higher-resolution native data set by molecular replacement using PHASER (*McCoy et al., 2007*). The rounds of final refinement in PHENIX and Coot used the 2.4 Å native data with only model-based phases. The electron density allowed unambiguous side chain assignments for all of the ParE2 toxin and the C-terminal half of the ParD2 antitoxin (residues 43–89), which contains the entire toxin-binding region. There was a significant amount of electron density that must correspond to the N-terminus of the antitoxin but it was mostly uninterpretable except for a noisy helical density feature that appears to be antitoxin residues 12–25. There are no protein-protein crystal contacts in this region and the protein seems to adopt multiple conformations. Given the high symmetry of the apparent space group of the crystal (I4122), the possible lower symmetries (P422, I4, I222) were explored in an attempt to account for this density without success. We believe that the high final $R_{free}$ for the refinement of this structure reflects our inability to model these noisy features corresponding to the flexible N-terminus of the antitoxin.

To build a second model for the ParD2-ParE2 interface, we co-crystallized ParE2 in complex with ParD2 modified to have the rigid, N-terminal 45 amino acids of ParD3 (ParD2*; ML3305). Importantly, neither ParD2 nor ParD3 contain any predicted toxin-binding residues in this N-terminal region. ParD2*-ParE2 was purified using the same general approach as ParD2-ParE2 and was stored in a buffer of 10 mM Tris-HCl pH 8.0 and 150 mM NaCl. Initial crystallization screening was carried out as before using 8 mg/mL of ParD2*-ParE2. The best crystal grew in condition 55 of the PegRx HT kit (0.2 M ammonium acetate, 0.1 M sodium citrate tribasic dihydrate pH 5.5, 24% v/v polyethylene glycol 400). Diffraction data was collected on the 24-ID-E beamline at APS (NE-CAT) to 2.9 Å resolution and indexed, integrated, and scaled using XDS. The structure was solved using molecular replacement with the native ParD2-ParE2 structure, and the model was further refined iteratively using PHENIX and Coot.

## Acknowledgements

We thank D Ding, C McClune, A Keating, and R Gaudet for comments on the manuscript. The project described was supported by award T32GM007753 from the National Institute of General Medical Sciences. M.T.L. is an Investigator of the Howard Hughes Medical Institute. This research made use of NE-CAT beamlines (P30 GM124165), a Pilatus detector (RR029205), and an Eiger detector (OD021527) at the APS (DE-AC02-06CH11357).

# Additional information

### Competing interests
Michael T Laub: Reviewing editor, *eLife*. The other authors declare that no competing interests exist.

### Funding

| Funder | Grant reference number | Author |
|---|---|---|
| National Institutes of Health | T32GM007753 | Thuy-Lan V Lite |
| Howard Hughes Medical Institute | | Michael T Laub |

The funders had no role in study design, data collection and interpretation, or the decision to submit the work for publication.

### Author contributions
Thuy-Lan V Lite, Conceptualization, Data curation, Formal analysis, Validation, Investigation, Visualization, Methodology, Writing - original draft, Writing - review and editing; Robert A Grant, Data curation, Formal analysis, Validation, Investigation, Visualization, Methodology; Isabel Nocedal, Monica S Guo, Formal analysis; Megan L Littlehale, Investigation, Methodology; Michael T Laub, Conceptualization, Data curation, Formal analysis, Supervision, Funding acquisition, Visualization, Writing - original draft, Project administration, Writing - review and editing

### Author ORCIDs
Thuy-Lan V Lite (iD) https://orcid.org/0000-0003-2743-4231
Isabel Nocedal (iD) http://orcid.org/0000-0002-4706-1113
Michael T Laub (iD) https://orcid.org/0000-0002-8288-7607

### Decision letter and Author response
Decision letter https://doi.org/10.7554/eLife.60924.sa1
Author response https://doi.org/10.7554/eLife.60924.sa2

# Additional files

### Supplementary files
• Source data 1. ParE protein tree.

• Source data 2. ParE protein alignment.

• Source data 3. Species tree.

• Source data 4. Species alignment.

• Source data 5. ParD2-ParE2 model.

• Source data 6. ParD2-ParE2 electron density.

• Supplementary file 1. Tables S1-S3. Table S1. Crystallographic data and refinement statistics Table S2. Plasmids created in this study Table S3. Oligonucleotides used in this study

• Transparent reporting form

### Data availability
Diffraction data have been deposited in PDB under the accession code 6X0A. Datasets generated during this study have been deposited in GEO. Raw data, variant frequency, and variant fitness scores can be found under the accession number GSE153897.

The following datasets were generated:

| | | Database and |
|---|---|---|

| Author(s) | Year | Dataset title | Dataset URL | Identifier |
|---|---|---|---|---|
| Lite TV, Grant RA, Laub MT | 2020 | X-ray structure of a chimeric ParDE toxin-antitoxin complex from Mesorhizobium opportunistum | https://www.rcsb.org/structure/6X0A | RCSB Protein Data Bank, 6X0A |
| Lite TV, Grant RA, Nocedal I, Guo MS, Laub MT | 2020 | The genetic landscape of protein-protein interaction specificity | https://www.ncbi.nlm.nih.gov/geo/query/acc.cgi?acc=GSE153897 | NCBI Gene Expression Omnibus, GSE153897 |

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
