## [Decision Letter]

**Acceptance summary:**

A cell is a very crowded environment that is prone to interaction promiscuity among proteins. Natural selection, therefore, acts to maintain interactions among cognate proteins but also to prevent the ones that may have deleterious consequences. Here, Lite and colleagues examine the contribution of single amino acid substitutions and their combinations to protein-protein interaction specificity using a toxin-antitoxin system. The results provide important insight into how protein-protein interactions evolve and achieve specificity in the context of gene duplication where duplicated proteins initially have the same interaction partners but eventually evolve to have their specific cognate partners.

**Decision letter after peer review:**

Thank you for submitting your article "The genetic landscape of protein-protein interaction specificity" for consideration by *eLife*. Your article has been reviewed by Olga Boudker as the Senior Editor, a Reviewing Editor, and three reviewers. The reviewers have opted to remain anonymous.

The reviewers have discussed the reviews with one another and the Reviewing Editor has drafted this decision to help you prepare a revised submission.

Summary:

Lite and colleagues examine the contribution of single amino acid substitutions and their combinations to protein-protein interactions using a toxin-antitoxin system. The results provide an important insight into how protein-protein interactions evolve and achieve specificity in the context of gene duplication where duplicated proteins initially have the same interaction partners but eventually evolve to have their specific cognate partners. The reviewers appreciated the quality and importance of the work. They raised several points that would need to be addressed.

1) The novelty of the present study relative to previous work by the same authors and others is not obvious. The context in which the study is anchored could be broader to better demonstrate the implications of the work. Some previous studies that relate to the present work are not discussed.

2) There are some issues regarding the way fitness is estimated across backgrounds and regarding the analysis of epistasis.

3) Many analyses are based on precise cut offs. It would be important to show that the conclusions are robust to the choice of cut off values. A more quantitative assessment of the results rather than one based on cut offs could be used.

4) Issues related to data availability were raised.

5) Protein abundance of the various mutants does not seem to be taken into account and this may be a confounding factor in the measurement of protein-protein interactions. Ideally this would be addressed experimentally but it should at least be discussed if experiments are not easily feasible in the current context or if such data is not already available.

I am leaving the individual reports appended below because they are largely non-redundant and some of the details could be useful for preparing the revisions.

Reviewer #1:

Lite and colleagues describe the contributions of individual residues on protein-protein interaction specificity in the parDE3 toxin-antitoxin system. How interaction specificity arises in PPIs, particularly after gene duplication, is an important problem in evolutionary. They use bulk competition assays, coupled with deep sequencing to quantify the degree to which mutant antitoxins can detoxify the presence of the cognate interaction partner ParE3 and a homologous, orthogonal toxin ParE2. Novel findings include a quantitative exploration of the degeneracy of the cognate parDE3 toxin-antitoxin interface towards change in the targeted residues, an assessment of the relative contributions of each selected residue towards antitoxin binding and selectivity, a structural basis for the interface differences between parDE2 and parDE3 and proof that not all positive elements serve as specificity-determining residues. The data presented in this study should be of interest to biochemists, evolutionary biologists, and geneticists, so is appropriate for *eLife*'s general audience.

Lite et al., characterize the detoxifcation-capabilities of an antitoxin library consisting of 8000 variants, in an approach similar to a previously published study from the same authors (Aarke et al., 2015). While the new library has the advantage of being combinatorically complete, it is smaller than the library from the cited study and 2/3 of the investigated residues in this work were already mutated in the preceding study (albeit with only 13 / 10 of possible amino acids) There is an interesting difference between the previous work, which only used naturally occurring states, whereas this study uses all possible mutations. I think it would be very interesting if the authors could comment on how this might relate to the different results between the studies.

The Results section of this study makes it sound as if the specificity-switching residues were first identified herein. Overall, the similarity to previously published data and experimental setup alongside a decrease in total library size compared to previous studies, that investigate the same model system, make the novelty of this work a little less obvious. Perhaps the authors could emphasize again conceptually (apart from saturation) what new approach this study adds, and why it is crucial for our understanding of specificity.

The assay employed in this work is not capable of discriminating good antitoxins from great antitoxins (as discussed in subsection “Specificity arises from the discrimination between cognate and non-cognate partners”). To address this, the authors measure the contributions to specificity of all single amino acid changes in all possible genetic backgrounds. Such a background-independent analysis leverages the completeness of the library (and I think was first employed here, Salinas and Ranganathan, 2018) to get an average effect for each mutation, though I am a little unsure I understand that it is appropriate in this case. The authors conclude that the naturally occurring specificity-determining residues are "optimal with respect to promoting the insulation" (Introduction). It seems to me that the authors compare the fitness of each mutation in the context of all genetic backgrounds to the wild-type state in its wild-type context and not in the context of all genetic backgrounds. It is thus to be expected that the apparent fitness of antitoxins that retain the cognate amino acid in a given position (e.g. ParD3 retains a D in position 1) is <1 if they are analyzed in a background-dependent manner (e.g. DXX). This is because many of the backgrounds will contain deleterious states. So the comparison of a background averaged fitness to the fitness of the WT state in a single background seems inappropriate to me to show that the WT sequence is optimal.

Reviewer #2:

In this work, Li et al., perform an exceptionally comprehensive assessment of how individual mutations contribute to recognition specificity amongst paralogous complexes. The authors take a bacterial anti-toxin, ParD3, and construct a combinatorial saturation mutagenesis library of three positions that physically interface with the toxin, ParE3. They then measure the effects of these mutations on: (1) binding of the cognate partner, ParE3 and (2) binding of a non-cognate partner, the paralog ParE2. The authors also compare the interactions at these three positions in the crystal structures of the ParE3/ParD3 complex (previously published) and the ParE2/ParD2 (which they solved in this work).

The central result of the paper is that individual positions can act as both positive and negative elements for interaction specificity, and that "positive and negative contributions are neither inherently coupled nor mutually exclusive". That is, rather than using distinct sets of positions to either enhance cognate interactions or discourage non-cognate binding, specificity can be (but is not always) encoded in an overlapping set of residues. This has implications for both the engineering and evolution of protein interactions. The experiments appear to be of high technical quality, and we expect the results will be of interest to a diverse scientific audience in biophysics, structural biology, protein engineering and molecular evolution. We recommend publication in *eLife*.

Essential revisions:

1) In Figure 1B, the authors lay out two different models of how positions at a physical interface contribute to the formation of a specific protein-protein interaction. Their data demonstrate that both models apply within a single protein. Much of the impact of the paper seems to depend on how surprising (or not) this finding is, and what the consequences of this finding might be both for evolving and engineering complexes. Thus, it is necessary for the authors to provide more context that can help the reader clearly assess the impact of this work. Specifically, can more be said about why and when one model would be favored over another? What is the evolutionary implication for individual residues in a protein to have negative and/or positive roles in identifying cognate interactions? Why is it surprising that residues can carry out dual roles in recognition and discrimination in a binding partner?

2) The linear model (Figure 3—figure supplement 1, Figure 4—figure supplement 2) indicates that the effects of mutations at the anti-toxin/toxin interface can be considered near-independently. This suggests that there is little epistasis (or coupling) between positions. However, can the authors perform a more thorough analysis of second and third order epistasis? Do they see that epistasis is centered at zero for the average interaction between mutations across a pair of sites? Given that the authors have all of the data, a thorough analysis of epistasis would further support their linear model and show that the contributions of each position studied in ParD3 are independent from each other. This seems especially interesting given the spatial proximity of positions 61 and 64.

3) As the authors point out, the in vivo assays rely on induced expression of toxins and antitoxins in a heterologous host. They claim that "the relative behavior of ParD3 variants measured here is likely to apply in whatever context they arise". Can the authors show that the induction conditions do not strongly affect their measurements? One way to demonstrate this is to take a small panel of ~10-20 ParD3 mutants that have a broad range of effects on growth rate across both ParE3 and ParE2 backgrounds. The growth rate effects of mutants in this "sub-library" can then be measured across varying concentrations of IPTG (for induction of ParD3) and arabinose (induction of ParE2/E3). Does the rank ordering of the mutant growth rates effects change with induction level, or are the results indeed qualitatively similar?

Reviewer #3:

This is an interesting study that uses a combinatorially complete deep mutagenesis strategy to identify determinants of the specificity of protein-protein interactions between bacterial toxins and antitoxins using a paralogy pair as a model system. I enjoyed the manuscript: it addresses a general and important question with a good experimental design and using an elegant model system. The clarification of negative and positive contributions to specificity is conceptually interesting.

Essential revisions:

1) Many of the analyses use an arbitrary cut-off of interaction vs no interaction (W>0.5). Whilst this simplifies communication and analyses, it is important to: (1) demonstrate that the conclusions are robust to the choice of this arbitrary cut-off; and (2) to stress that this reduction in fitness is huge and in natural populations much smaller changes in fitness are likely to be selected against, especially in microbes with large effective population sizes. How does imposing a much higher fitness cut-off alter the authors' conclusions?

2) In general, and related to point [1], I would prefer to see a more quantitative treatment of the data. Defining proteins as interacting or not interacting is a bit clunky given that binding is actually a fully quantitative trait and the data here is, at least by design, also quantitative. Many of the questions addressed could be answered quantitatively rather than by using a binary categorisation of binding vs non-binding.

3) What about protein abundance? The authors don't quantify the effects of the combinatorial mutations on protein abundance, so some of the non-specific effects on binding are likely to be due to changes in concentration of the protein not the binding affinity.

4) Relationship to conclusions in previous publications from the same group: previous manuscripts from the same lab have focussed on the finding that mutational effects in protein interaction interfaces change in the presence of additional mutations in the same protein (Podgornaia and Laub, 2015 i.e. the importance of pairwise and higher order epistasis). How, quantitatively, does the current dataset compare to this previous dataset on a different system and also to previous toxin-anti toxin mutagenesis datasets? Are mutational effects less background dependent in the toxin-anti-toxin system or similarly so?

5) "Although a purely additive model of residue contributions was highly predictive of variant fitness (R2 = 0.89, SD between folds + 0.003; Figure 3—figure supplement 1B), the model was weakest for the most fit variants, likely due to diminishing returns for highly favorable residues." This is fully expected- mutations should have effects that are additive for free energy changes (ddG binding) but not for changes in protein concentration because of the non-linear relationship between the amount of protein bound to an interaction partner and the free energy of binding (dG).

6) Data processing / quality control. We tried to download the raw sequencing data from SRA using the referee token to perform some basic quality control, but it seems only possible to obtain the summary tables not the actual sequence reads (this may be an issue in general with private SRA entries). Would it be possible to get access to the raw data e.g. by making the entry fully public prior to publication for reviewing purposes?

We have noticed major issues with quite a few deep mutational scanning data analyses, including in published papers. These include inappropriate filtering such that the analysed datasets consist partly (or in some cases, largely) of sequencing errors not real variants which can seriously alter conclusions. A second issue is underestimating sampling errors due to over-sequencing. For the design of the library used here and the filtering applied, this seem unlikely to be the case. But it would still be good to run some basic checks prior to publication.

7) Introduction/ citation of prior work. An obvious missing citation given the similarity of questions, strategy, title and journal is: Diss et al., 2018. In addition, there have now been quite a few papers published that use a similar combinatorially complete deep mutagenesis design published on different molecules (proteins, RNAs) and molecular processes and very few of them are cited here. Also, the statement in the introduction that 'prior work has not used combinatorially complete libraries to systematically dissect interface residues' is a bit misleading given previous work from the Laub lab.

8) It would be useful to more explicitly state in the text the (rational for the) previous ParD mutagenesis library design in the previous publication by the same lab.

[Editors' note: further revisions were suggested prior to acceptance, as described below.]

Thank you for resubmitting your work entitled "Uncovering the basis of protein-protein interaction specificity with a combinatorially complete library" for further consideration by *eLife*. Your revised article has been evaluated by Olga Boudker (Senior Editor), a Reviewing Editor and one of the original reviewers.

The manuscript has been improved but there are some minor remaining issues that need to be addressed before formal acceptance, as outlined below:

I would include some of the panels from Figure 2—figure supplement 2 in the main figure – as the interaction strength increases, the degeneracy of the interface decreases. I think this is an interesting point.

Figure 4C, D. It looks like, if the changes in mutational effects across genetic backgrounds are significant, that there is more sign epistasis (switches from beneficial to detrimental mutational effects) for interaction with the non-cognate partner. Is this true? Any ideas why this might be?

Figure 3—figure supplement 1D. There is a (weak) sigmoidal relationship between the fitness scores predicted by the linear model and the actual fitness scores which suggests there is global (non-specific) epistasis in this system. This may be because there is a non-linear relationship between changes in free energy and the phenotype being measured (=~binding), as may be expected from thermodynamics. And/or perhaps because of an upper or lower bound on the measurement range.

---

## [Author Response]

Reviewer #1:Lite and colleagues describe the contributions of individual residues on protein-protein interaction specificity in the parDE3 toxin-antitoxin system. How interaction specificity arises in PPIs, particularly after gene duplication, is an important problem in evolutionary. They use bulk competition assays, coupled with deep sequencing to quantify the degree to which mutant antitoxins can detoxify the presence of the cognate interaction partner ParE3 and a homologous, orthogonal toxin ParE2. Novel findings include a quantitative exploration of the degeneracy of the cognate parDE3 toxin-antitoxin interface towards change in the targeted residues, an assessment of the relative contributions of each selected residue towards antitoxin binding and selectivity, a structural basis for the interface differences between parDE2 and parDE3 and proof that not all positive elements serve as specificity-determining residues. The data presented in this study should be of interest to biochemists, evolutionary biologists, and geneticists, so is appropriate for eLife's general audience.Lite et al., characterize the detoxifcation-capabilities of an antitoxin library consisting of 8000 variants, in an approach similar to a previously published study from the same authors (Aarke et al., 2015). While the new library has the advantage of being combinatorically complete, it is smaller than the library from the cited study and 2/3 of the investigated residues in this work were already mutated in the preceding study (albeit with only 13 / 10 of possible amino acids) There is an interesting difference between the previous work, which only used naturally occurring states, whereas this study uses all possible mutations. I think it would be very interesting if the authors could comment on how this might relate to the different results between the studies.The Results section of this study makes it sound as if the specificity-switching residues were first identified herein. Overall, the similarity to previously published data and experimental setup alongside a decrease in total library size compared to previous studies, that investigate the same model system, make the novelty of this work a little less obvious. Perhaps the authors could emphasize again conceptually (apart from saturation) what new approach this study adds, and why it is crucial for our understanding of specificity.

We have now substantially revised the Introduction to provide more context for our study. In particular, we now better describe a series of prior deep-mutational scanning studies and how the work at hand represents a conceptual advance over these prior studies and how the design of our experiment enables new insights into protein interaction *specificity*, not just binding of a cognate partner.

We also now emphasize the difference between the library constructed here and that constructed previously. The new library is (i) focused on a more appropriate set of interface residues and (ii) is combinatorially complete, which ultimately enables much more powerful and rigorous conclusions. To expand on this last point: in principle, the prior ParD3 library could be used to perform some of the same analyses performed here that investigate how individual substitutions impact interaction specificity in multiple genetic contexts. However, such analyses were not reported in the prior study of ParD3 and would be based on a much more limited number of contexts for virtually all substitutions. We have now run the same analyses presented in Figure 4C-E, but using only the subset of the library that overlaps with what we used previously in Aakre et al., 2015. A direct comparison of the modes calculated for each library clearly demonstrates that the prior library would not have produced the same results as those generated here using the combinatorially complete library. This analysis is now included in Figure 4—figure supplement 3A-D.

The assay employed in this work is not capable of discriminating good antitoxins from great antitoxins (as discussed in subsection “Specificity arises from the discrimination between cognate and non-cognate partners”). To address this, the authors measure the contributions to specificity of all single amino acid changes in all possible genetic backgrounds. Such a background-independent analysis leverages the completeness of the library (and I think was first employed here, Salinas and Ranganathan, 2018) to get an average effect for each mutation, though I am a little unsure I understand that it is appropriate in this case. The authors conclude that the naturally occurring specificity-determining residues are "optimal with respect to promoting the insulation" (Introduction). It seems to me that the authors compare the fitness of each mutation in the context of all genetic backgrounds to the wild-type state in its wild-type context and not in the context of all genetic backgrounds. It is thus to be expected that the apparent fitness of antitoxins that retain the cognate amino acid in a given position (e.g. ParD3 retains a D in position 1) is <1 if they are analyzed in a background-dependent manner (e.g. DXX). This is because many of the backgrounds will contain deleterious states. So the comparison of a background averaged fitness to the fitness of the WT state in a single background seems inappropriate to me to show that the WT sequence is optimal.

We have now directly addressed this concern by performing a similar analysis to Figure 4, but comparing fitness distributions (*W*, not ∆*W*) for each amino acid at each position to the fitness distribution for the corresponding wild-type residue. In other words, we compare AXX, CXX, EXX, etc. distributions to the distribution featuring the wild-type residue at position 1, i.e. DXX. This analysis (see revised Figure 4—figure supplement 3E-G) produced highly similar results to those shown in Figure 4, supporting the notion that the wild-type sequence is optimal.

Reviewer #2:In this work, Li et al., perform an exceptionally comprehensive assessment of how individual mutations contribute to recognition specificity amongst paralogous complexes. The authors take a bacterial anti-toxin, ParD3, and construct a combinatorial saturation mutagenesis library of three positions that physically interface with the toxin, ParE3. They then measure the effects of these mutations on: (1) binding of the cognate partner, ParE3 and (2) binding of a non-cognate partner, the paralog ParE2. The authors also compare the interactions at these three positions in the crystal structures of the ParE3/ParD3 complex (previously published) and the ParE2/ParD2 (which they solved in this work).The central result of the paper is that individual positions can act as both positive and negative elements for interaction specificity, and that "positive and negative contributions are neither inherently coupled nor mutually exclusive". That is, rather than using distinct sets of positions to either enhance cognate interactions or discourage non-cognate binding, specificity can be (but is not always) encoded in an overlapping set of residues. This has implications for both the engineering and evolution of protein interactions. The experiments appear to be of high technical quality, and we expect the results will be of interest to a diverse scientific audience in biophysics, structural biology, protein engineering and molecular evolution. We recommend publication in eLife.Essential revisions:1) In Figure 1B, the authors lay out two different models of how positions at a physical interface contribute to the formation of a specific protein-protein interaction. Their data demonstrate that both models apply within a single protein. Much of the impact of the paper seems to depend on how surprising (or not) this finding is, and what the consequences of this finding might be both for evolving and engineering complexes. Thus, it is necessary for the authors to provide more context that can help the reader clearly assess the impact of this work. Specifically, can more be said about why and when one model would be favored over another? What is the evolutionary implication for individual residues in a protein to have negative and/or positive roles in identifying cognate interactions? Why is it surprising that residues can carry out dual roles in recognition and discrimination in a binding partner?

As we note in the Introduction, it is not yet clear for virtually all protein complexes how specificity is determined and whether individual residues play positive or negative roles, or both. Thus, we did not approach this study with an a priori notion of what would constitute 'surprising' results and we did not want to write the Introduction in too leading a way. Instead, we wanted to indicate that we simply don't know the answer for most protein complexes as there have been exceedingly few studies like ours using combinatorially complete libraries that can properly address this question. As noted above in response to reviewer 1, we have now revised the Introduction to better lay out what prior studies have shown and how our combinatorially complete library approach differs, enabling new insights into protein interaction specificity. We hope these revisions now provide the context this reviewer was seeking. Additionally, based on this reviewer's comment we have also amended the Introduction to highlight the fact that systematically dissecting interaction specificity has implications both for understanding protein evolution and for protein engineering efforts.

2) The linear model (Figure 3—figure supplement 1, Figure 4—figure supplement 2) indicates that the effects of mutations at the anti-toxin/toxin interface can be considered near-independently. This suggests that there is little epistasis (or coupling) between positions. However, can the authors perform a more thorough analysis of second and third order epistasis? Do they see that epistasis is centered at zero for the average interaction between mutations across a pair of sites? Given that the authors have all of the data, a thorough analysis of epistasis would further support their linear model and show that the contributions of each position studied in ParD3 are independent from each other. This seems especially interesting given the spatial proximity of positions 61 and 64.

We have now examined epistasis in more detail, following an approach similar to that presented in Salinas and Ranganathan, (2018). We calculated the difference between the fitness of each variant predicted assuming independence and the observed fitness. We then plotted these distributions for all possible double mutants (broken down by position pair) and all possible triple mutants. As expected given the success of our linear model, the distribution of differences in each case is centered around 0 indicating that, on average, the two positions behave largely independently, i.e. there is minimal epistasis. This new analysis is presented in subsection “Library variants show a range of abilities to discriminate between cognate and non-cognate partners” and in Figure 3—figure supplement 1E-F of the revised manuscript. We also note in the text that the lack of epistasis in our library contrasts with that observed previously by Salinas and Ranganathan in PDZ-peptide interactions and by us in the context of two-component signaling proteins. However, the library constructed here involves three positions in ParD that interact with non-overlapping sets of residues in ParE (even for positions 61 and 64 noted by the reviewer), potentially explaining their independence.

3) As the authors point out, the in vivo assays rely on induced expression of toxins and antitoxins in a heterologous host. They claim that "the relative behavior of ParD3 variants measured here is likely to apply in whatever context they arise". Can the authors show that the induction conditions do not strongly affect their measurements? One way to demonstrate this is to take a small panel of ~10-20 ParD3 mutants that have a broad range of effects on growth rate across both ParE3 and ParE2 backgrounds. The growth rate effects of mutants in this "sub-library" can then be measured across varying concentrations of IPTG (for induction of ParD3) and arabinose (induction of ParE2/E3). Does the rank ordering of the mutant growth rates effects change with induction level, or are the results indeed qualitatively similar?

We have now performed this analysis. Given restrictions on lab time during the pandemic and the first author’s return to medical school (she is an MD-PhD student), we used a smaller (4 antitoxin variants) panel than suggested by the reviewer as this subset of clones already existed (Figure 1—figure supplement 1E) and did not require de novo construction/cloning. Using this panel, we plated strains harboring each antitoxin variant and either the ParE3 or ParE2 toxin on varying concentrations of IPTG (the antitoxin inducer). As is clear in the new Figure 2—figure supplement 3 the rank ordering of mutant growth rates is unaffected by induction level.

Reviewer #3:This is an interesting study that uses a combinatorially complete deep mutagenesis strategy to identify determinants of the specificity of protein-protein interactions between bacterial toxins and antitoxins using a paralogy pair as a model system. I enjoyed the manuscript: it addresses a general and important question with a good experimental design and using an elegant model system. The clarification of negative and positive contributions to specificity is conceptually interesting.Essential revisions:1) Many of the analyses use an arbitrary cut-off of interaction vs no interaction (W>0.5). Whilst this simplifies communication and analyses, it is important to: (1) demonstrate that the conclusions are robust to the choice of this arbitrary cut-off; and (2) to stress that this reduction in fitness is huge and in natural populations much smaller changes in fitness are likely to be selected against, especially in microbes with large effective population sizes. How does imposing a much higher fitness cut-off alter the authors' conclusions?

There are actually relatively few analyses in the paper that rely on an arbitrary cut-off. For instance, the data in Figure 4 involve the presentation of the full distribution of fitness effects, with no thresholds or cut-offs used beyond the exclusion of proline substitutions that may trivially disrupt protein structure. We do introduce thresholds in the analyses shown in Figure 2D, Figure 2G, and Figure 3B-C in which we present sequence logos for ParD variants that are ParE3-specific, ParE2-specific, or promiscuous. However, we also included sequence logos for each category at several different thresholds (0.5, 0.7, and 0.9) in Figure 2—figure supplement 2. We included these for precisely the reason raised by the reviewer, namely to show that the conclusions drawn are robust to the choice of threshold. We have now added statements (see subsection “Pervasive degeneracy in toxin binding” the revised manuscript) drawing the reader's attention to Figure 2—figure supplement 2 and have stressed (see the Discussion section) the point that large population sizes for microbes can lead to selection on fitness differences that are difficult to detect in a laboratory setting.

2) In general, and related to point [1], I would prefer to see a more quantitative treatment of the data. Defining proteins as interacting or not interacting is a bit clunky given that binding is actually a fully quantitative trait and the data here is, at least by design, also quantitative. Many of the questions addressed could be answered quantitatively rather than by using a binary categorisation of binding vs non-binding.

Please see the immediately preceding response on the same topic.

3) What about protein abundance? The authors don't quantify the effects of the combinatorial mutations on protein abundance, so some of the non-specific effects on binding are likely to be due to changes in concentration of the protein not the binding affinity.

Each of the positions mutated in our library are solvent-exposed residues that show high variability in naturally occurring ParD homologs. Thus, we think it is unlikely that the mutations introduced affect protein folding/stability and, consequently, abundance, but we cannot formally rule this out. There are no antibodies to ParD available and epitope-tagging ParD can interfere with function. Thus, we have not been able to experimentally address this issue, but we do now explicitly discuss it in the revised manuscript (see the Discussion section).

4) Relationship to conclusions in previous publications from the same group: previous manuscripts from the same lab have focussed on the finding that mutational effects in protein interaction interfaces change in the presence of additional mutations in the same protein (Podgornaia and Laub, 2015 i.e. the importance of pairwise and higher order epistasis). How, quantitatively, does the current dataset compare to this previous dataset on a different system and also to previous toxin-anti toxin mutagenesis datasets? Are mutational effects less background dependent in the toxin-anti-toxin system or similarly so?

Please see our response to reviewer 2, point #2 on the same issue.

5) "Although a purely additive model of residue contributions was highly predictive of variant fitness (R2 = 0.89, SD between folds + 0.003; Figure 3—figure supplement 1B), the model was weakest for the most fit variants, likely due to diminishing returns for highly favorable residues." This is fully expected- mutations should have effects that are additive for free energy changes (ddG binding) but not for changes in protein concentration because of the non-linear relationship between the amount of protein bound to an interaction partner and the free energy of binding (dG).

We agree and have added a statement in subsection “Library variants show a range of abilities to discriminate between cognate and non-cognate partners” indicating that our model is consistent with mutations having strictly additive effects on binding.

6) Data processing / quality control. We tried to download the raw sequencing data from SRA using the referee token to perform some basic quality control, but it seems only possible to obtain the summary tables not the actual sequence reads (this may be an issue in general with private SRA entries). Would it be possible to get access to the raw data e.g. by making the entry fully public prior to publication for reviewing purposes?We have noticed major issues with quite a few deep mutational scanning data analyses, including in published papers. These include inappropriate filtering such that the analysed datasets consist partly (or in some cases, largely) of sequencing errors not real variants which can seriously alter conclusions. A second issue is underestimating sampling errors due to over-sequencing. For the design of the library used here and the filtering applied, this seem unlikely to be the case. But it would still be good to run some basic checks prior to publication.

We have confirmed that the SRA data is complete and available. To avoid any further issues, we have made everything publicly available in advance of publication – please see https://www.ncbi.nlm.nih.gov/sra?term=SRP270411. If the reviewers wish to download the raw data and find any issues in their own quality control, we are certainly happy to hear about it and address it. Please note that we detected contamination in one post-selection library sample with the wild-type ParD3 gene (<0.4% of aligned reads for that sample), which was filtered out. This wild-type sequence was likely introduced during library preparation because (1) the wild-type plasmid was not detected in the pre-selection sample, (2) the wild-type plasmid was not used in the construction of the library (see Supplementary file 2), and (3) the wild-type plasmid could not have been created during library construction because it does not strictly use NNK codons at the mutated positions. The exact filtering steps we used (i.e., requiring NNK codons at mutated positions and 0 mutations at all remaining positions) are indicated in the Materials and methods section.

7] Introduction/ citation of prior work. An obvious missing citation given the similarity of questions, strategy, title and journal is: Diss et al., 2018. In addition, there have now been quite a few papers published that use a similar combinatorially complete deep mutagenesis design published on different molecules (proteins, RNAs) and molecular processes and very few of them are cited here. Also, the statement in the introduction that 'prior work has not used combinatorially complete libraries to systematically dissect interface residues' is a bit misleading given previous work from the Laub lab.

As noted above, we have now expanded and revised the Introduction quite substantially to better cite and discuss prior deep mutational scans, including the Diss and Lehner study noted by the reviewer and our own lab's work, with an emphasis on how our study builds off but differs from those prior studies.

8] It would be useful to more explicitly state in the text the (rational for the) previous ParD mutagenesis library design in the previous publication by the same lab.

We now provide additional discussion in the text (see the Results section) of the rationale for the previous ParD library, emphasizing the modifications made in this work and their advantages.

[Editors' note: further revisions were suggested prior to acceptance, as described below.]

I would include some of the panels from Figure 2—figure supplement 2 in the main figure – as the interaction strength increases, the degeneracy of the interface decreases. I think this is an interesting point.

We have now moved all of the panels from former Figure 2—figure supplement 2 into the main figure.

Figure 4C,D. It looks like, if the changes in mutational effects across genetic backgrounds are significant, that there is more sign epistasis (switches from beneficial to detrimental mutational effects) for interaction with the non-cognate partner. Is this true? Any ideas why this might be?

We agree with the reviewer's point that the distributions in Figure 4C-D can extend above and below 0, especially for the non-cognate interaction, which suggests that a mutation can have either beneficial or detrimental effects depending on the exact context. If they had been bimodal with a peak above 0 and one below 0, we think it would have been important to comment on such sign epistasis. However, because these distributions are unimodal, we think it would be overly speculative to comment in depth on the issue of sign epistasis. Furthermore, for 4D panel 1 (first library position), where this phenomenon is most common, the distributions of fitness effects are tightly centered around 0. The complete data are, of course, provided with the paper and readers will be able to dissect this aspect of our data, if they so desire.

Figure 3—figure supplement 1D. There is a (weak) sigmoidal relationship between the fitness scores predicted by the linear model and the actual fitness scores which suggests there is global (non-specific) epistasis in this system. This may be because there is a non-linear relationship between changes in free energy and the phenotype being measured (=~binding), as may be expected from thermodynamics. And/or perhaps because of an upper or lower bound on the measurement range.

We agree and had already commented on this in the Discussion section. We have now added a similar comment in the Results section when Figure 3—figure supplement 1D is first cited.